# Sterols are required for the coordinated assembly of lipid droplets in developing seeds

Linhui Yu[1], Jilian Fan [1], Chao Zhou[1] & Changcheng Xu [1✉]

Lipid droplets (LDs) are intracellular organelles critical for energy storage and lipid metabolism. They are typically composed of an oil core coated by a monolayer of phospholipids and proteins such as oleosins. The mechanistic details of LD biogenesis remain poorly defined. However, emerging evidence suggest that their formation is a spatiotemporally regulated process, occurring at specific sites of the endoplasmic reticulum defined by a specific set of lipids and proteins. Here, we show that sterols are required for formation of oleosin-coated LDs in Arabidopsis. Analysis of sterol pathway mutants revealed that deficiency in several $\Delta^5$-sterols accounts for the phenotype. Importantly, mutants deficient in these sterols also display reduced LD number, increased LD size and reduced oil content in seeds. Collectively, our data reveal a role of sterols in coordinating the synthesis of oil and oleosins and their assembly into LDs, highlighting the importance of membrane lipids in regulating LD biogenesis.

[1] Biology Department, Brookhaven National Laboratory, Upton, NY 11973, USA. ✉email: cxu@bnl.gov

Sterols are isoprenoid-derived lipids with essential roles in cell structure, function, and physiology. As important components of biological membranes, sterols interact with phospholipids and proteins within the membrane, thereby regulating membrane fluidity, permeability, and membrane protein functions[1–3]. In addition to their structural role, sterols serve as a metabolic precursor to brassinosteroids (BRs), a group of steroid hormones that regulate a wide range of biological process[4]. Moreover, some sterols and sterol biosynthetic intermediates have been suggested to function as signaling molecules influencing transcription and signaling transduction pathways independent of BRs[5,6]. Furthermore, sterols are crucial players at the subcellular level in defining lipid rafts, sterol- and sphingolipid-enriched functional membrane microdomains that are present in diverse organisms[7]. Such structures have been implicated in numerous biological processes as diverse as immunity in plants[8], the synthesis of phospholipids in mammals[9] and the biogenesis of lipid droplets (LDs)[10,11], specialized subcellular organelles for energy storage and lipid metabolism in all eukaryotes.

Plant sterols are synthesized in the endoplasmic reticulum (ER) from acetyl-CoA via the mevalonate pathway leading to the formation of cycloartenol (Supplementary Fig. 1), the first parental sterol composed of a tetracyclic ring nucleus with a free hydroxyl group and side chain at the C-3 or C-24 carbon atom, respectively. Cycloartenol is the common substrate for the synthesis of cholesterol, a minor sterol in plants, and of the major sterols, sitosterol, stigmasterol, and campesterol. The conversion of cycloartenol into major plant sterols involves a series of enzymatic reactions including methylation, demethylation, reduction, isomerization, and desaturation[2,12]. Among these enzymes, sterol methyltransferase 2/3 (SMT2/3) and sterol methyl oxidase 2 (SMO2) define the second branch point in sterol biosynthesis, which directs precursor flow towards two parallel pathways leading to the generation of sitosterol and stigmasterol, two 24-ethyl-sterols or campesterol, a 24-methyl-sterol, respectively by both unique and shared enzymes, with the latter, including DWARF1 (DWF1), DWF5 and DWF7. Campesterol is the precursor to BRs. Consequently, disruption of the shared enzymatic steps downstream of SMT2/3 and SMO2 in the sterol biosynthetic pathway often leads to campesterol depletion, causing typical phenotypes of BR-deficient mutants such as severe dwarfism, impaired fertility, and yield. In addition to their free forms, sterols can be esterified with fatty acids to produce sterol esters (SEs) by sterol ester synthases, including phospholipid sterol acyltransferase1 (PSAT1) in Arabidopsis[13] and the resulting SEs are stored in LDs[2].

The physical structure of the LD consists of a hydrophobic core of SEs and triacyclglycerols (TAGs) surrounded by a monolayer of phospholipids and free sterols with a small set of proteins embedded or peripherally associated. Currently, the exact mechanistic details underlying LD biogenesis remain poorly understood[14,15]. The most widely accepted model proposes that LDs originate from the specific ER subdomains defined by a specific set of lipids such as diacylglycerol (DAG) and proteins[16–18], and is driven by ER membrane curvature, tension, and asymmetry[19,20]. Several ER membrane proteins have been implicated in LD biogenesis in yeast and animals, likely via their effects on local lipid composition and hence ER membrane environments at sites of LD formation[21–23]. Furthermore, abundant LD structural proteins such as oleosins, have been shown to play a role in LD formation by sequestering neutral lipids from the ER bilayer into LDs[24].

Oleosin possesses a long hydrophobic hairpin flanked by short amphipathic N- and C-terminal regions. On a LD, the hydrophobic hairpin is predicted to penetrate the monolayer into the neutral lipid core, whereas the flanking sequences interact with the monolayer membrane lipids on the LD surface[25]. During LD formation, oleosins are cotranslationally inserted into the ER, from where they then move to LDs. The Arabidopsis genome contains 17 oleosin genes specifically expressed in seeds and flower organs. Gene knockout or downregulation studies have revealed a key role of oleosins in regulating the size and stability of LDs in seeds[26,27], but the precise role of oleosins in LD biogenesis and dynamics remains largely unknown.

Studies in animal model systems have shown that sterol synthesis, TAG metabolism, LD accumulation, and phospholipid synthesis are coregulated by a family of transcription factors, sterol regulatory element-binding proteins to maintain intracellular lipid homeostasis[28]. In plants, genes encoding key regulatory enzymes in sterol biosynthetic pathways show similar expression patterns to those in fatty acid and TAG synthesis during embryo development (Supplementary Fig. 2). In addition, several enzymes involved in sterol synthesis have been shown to be coordinately regulated during oil-seed development[29] and the accumulation of high levels of free sterols precede the accumulation of TAGs in oilseeds[29,30], but the functional role of sterols in TAG storage has not been examined in plants.

To understand the mechanisms and factors controlling LD biogenesis, we carried out a forward genetic screen using a transgenic line constitutively overproducing a green fluorescent protein (GFP)-tagged OLEOSIN1 fusion protein (OLE1)[31] and searched for mutants with the altered size, number, and morphology of leaf LDs. We report here the identification and characterization of one of the mutants, M1-7, that is severely defective in the accumulation of LDs in leaves. We showed that altered membrane sterol composition, but not deficiency in BRs and SEs compromises LD accumulation in leaves and seeds. The possible implication of this finding in our understanding of the role of sterols in LD formation is discussed.

## Results

**Isolation of LD mutants**. Seeds of the homozygous Arabidopsis transgenic line overexpressing *OLE1* driven by the Cauliflower Mosaic Virus (CaMV) promoter[31] were mutagenized with ethyl methanesulfonate. To identify mutants defective in TAG accumulation, leaf samples from individual $M_2$ plants were examined under a fluorescence microscope. Among 10 putative mutants isolated thus far, one of the mutants, designated *M1-7*, was almost completely devoid of leaf LDs and TAG. Mutant plants grown on soil displayed dark-green leaves and a short, robust statute (Fig. 1a), and contained only a few LDs and very limited amounts of TAG in young and mature leaves, though both the TAG content and LD size were slightly increased as leaves aged (Fig. 1b–d). This mutant was subsequently backcrossed to the parental line. The self-pollinated $F_1$ plants produced $F_2$ progeny that segregated for dwarf plants in a ratio of ~4–1 (48 out of 205, 23.4%), and all the 48 plants had drastically reduced LD accumulation in leaves. These results suggest that LD deficiency and the dwarf phenotype are tightly linked and cosegregate as a single recessive trait.

**The *M1-7* mutant represents a new allele of *dwf*5**. To identify the mutation responsible for the LD-deficient phenotype, the *M1-7* mutant was outcrossed to ecotype Landsberg *erecta* to generate a mapping population that was used for rough mapping, and the mutant locus was delimited to an ~1500-kb region (Fig. 2a). Whole genomic DNA sequencing identified six GC:AT transitions that generated nonsense or missense mutations in protein-coding sequences in this region (Supplementary data 1). Among them is a G to A nonsense transition in the 11th exon of the gene At1g50430,

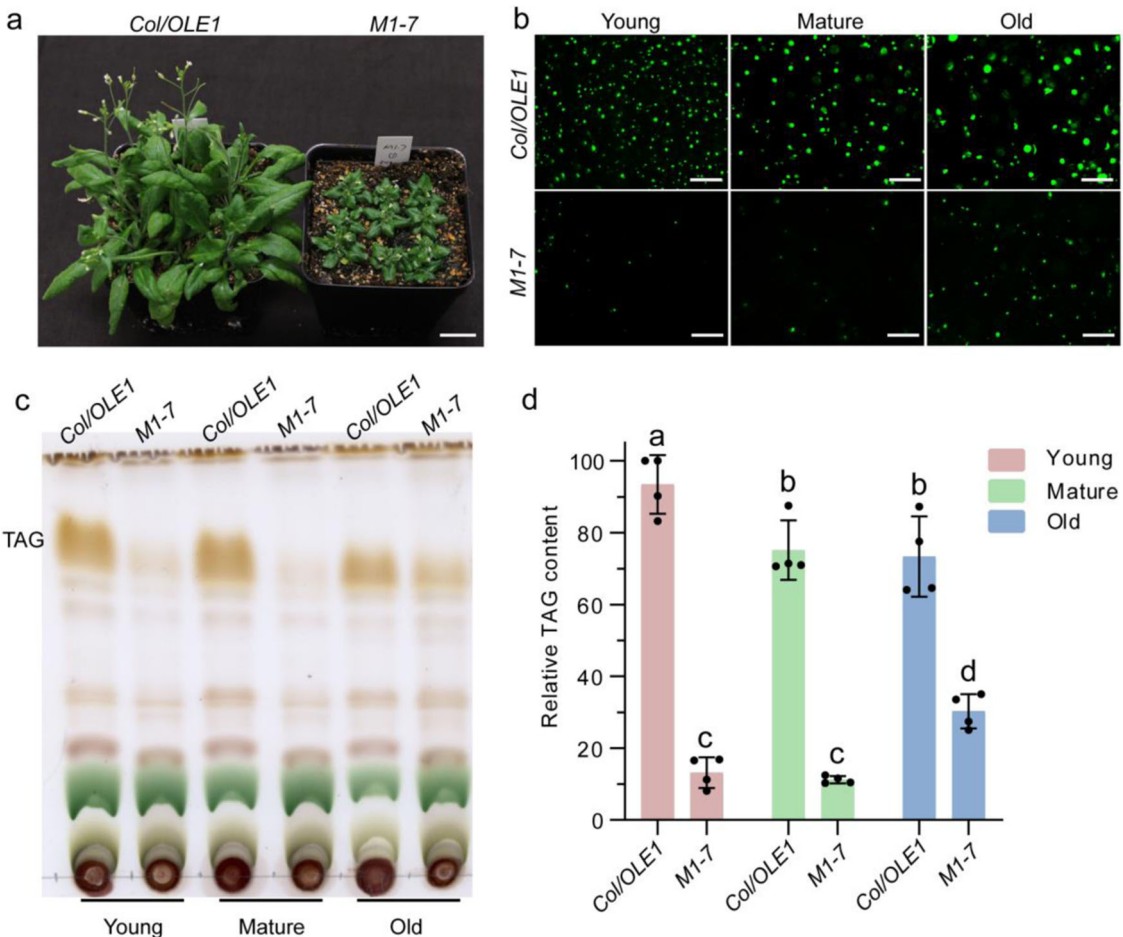

**Fig. 1 TAG content and LD abundance are dramatically decreased in *M1-7* mutant. a** Phenotypes of 4-week-old *M1-7* and *Col/OLE1*. Bar = 2 cm.
**b** Representative images of LDs (OLE1-GFP, green) in rosette leaves of different ages of 4-week-old plants (young leaves: the upper growing leaves; mature leaves: the newest expanded leaves; old leaves: the oldest expanded green leaves). The experiment was repeated 5 times with similar results. Bar = 50 μm.
**c** Thin-layer chromatogram of neutral lipids in leaves of *M1-7* and *Col/OLE1*. Lipids were visualized with 5% sulfuric acid by charring. **d** Relative TAG contents in leaf of *M1-7* and *Col/OLE1* quantified by Image J software. Data are mean ± SD. of four biological replicates. Different letters indicate significant differences at $P < 0.05$, as determined by one-way ANOVA with Tukey's multiple comparisons test.

which encodes DWF5, a sterol $\Delta^7$-sterol-C5-desaturase. This mutation introduces a premature stop codon that is expected to lead to a truncated protein lacking 76 amino acids at the C-terminus, and the *M1-7* mutant was hence renamed as *dwf5-10/OLE1* (Fig. 2b). To confirm that *DWF5* is the causative gene for the LD-deficient phenotype, we isolated a homozygous dwarf mutant with a T-DNA insertion in the 8th intron of the gene and designated it *dwf5-8* (Fig. 2a, b). This T-DNA mutant, when introduced into the *OLE1* line, recapitulated the neutral lipid storage phenotype of *dwf5-10/OLE1* (Fig. 2c–e). In addition, introducing the *DWF5* coding sequence driven with CaMV35S promoter into *dwf5-10/OLE1* led to increases in plant growth (Supplementary Fig. 3a, b) and LD abundance (Supplementary Fig. 3c).

**The role of sterols in neutral lipid storage is BR-independent.** To exclude the possibility that the LD-deficient phenotype was due to a lack of BRs or block of BR signaling, we isolated T-DNA insertion mutants in *DWF4* and *Brassinosteroid-Insensitive 1* (*BRI1*), which encodes an enzyme necessary for BR synthesis (Supplementary Fig. 1) and the BR receptor, respectively[4]. These two mutants, when introduced into the *OLE1* line, caused no obvious changes in leaf LD abundance and TAG content (Supplementary Fig. 4a, b). In addition, external feeding of 24-epibrassinolide (24-epiBL) failed to rescue the LD deficiency

in *dwf5-8/OLE1* and *dwf7-4/OLE1* (Supplementary Fig. 4c, d). Moreover, blocking endogenous BR biosynthesis by brassinazole (BRZ) or applying exogenous 24-epiBL has no effect on TAG content in shoots (Supplementary Fig. 4e). These data suggest that the LD- and TAG-deficient phenotype is not due to defects in BR synthesis or BR signaling transduction.

**Deficiency in 24-ethyl/ethylidene-$\Delta^5$-sterols results in decreases in leaf TAG accumulation in the *OLE1* line.** Sterol analysis showed that total free sterols decreased by approximately 74% in *dwf5* mutants compared with the wild type (WT), due primarily to drastic decreases in the major sterols, such as sitosterol, campesterol, stigmasterol, and isofucosterol (24-ethylidene-$\Delta^5$-sterol) (Supplementary Table 1). In addition, there were increases in levels of several unusual sterol molecules, such as stigmasta-5,7,22-trienol and stigmasta-5,7-dienol in *dwf5* mutants. To gain further insight into whether and how changes in sterol profiles impact TAG storage, we isolated several additional T-DNA insertion mutants defective in various steps of the sterol biosynthesis pathway and introduced these mutations into the *OLE1* line by genetic crossing. The *dwf1-2* mutant is defective in the conversion of 24-methylenecholesterol and isofucosterol to campesterol and sitosterol, respectively[32]. Consequently, this mutant lacks sitosterol and campesterol, but instead accumulates

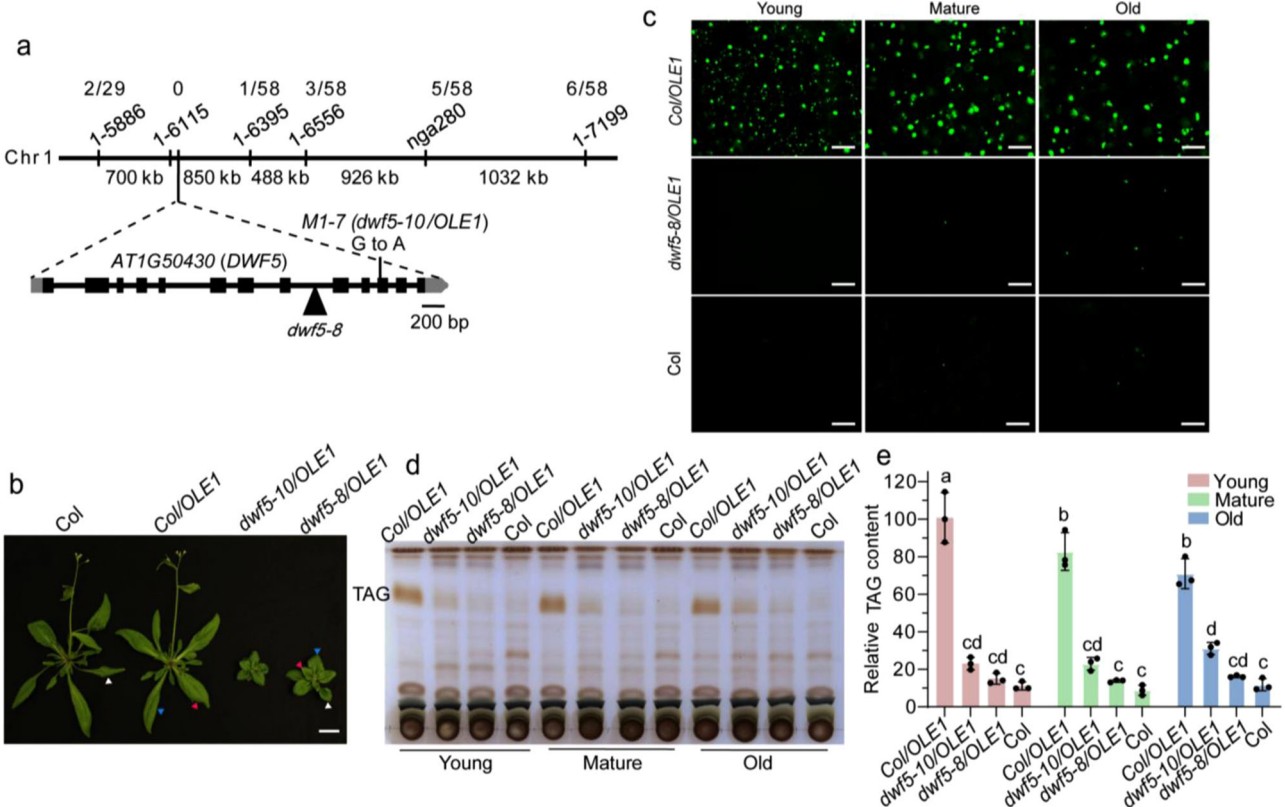

**Fig. 2 Disruption of DWF5 results in defects in TAG and LD accumulation in leaves. a** The map position of the *M1-7* mutation and structure of *DWF5* (At1g50430). Markers used for mapping and the respective number of recombinants are indicated. The exons of *DWF5* are shown as black boxes and introns are indicated by lines. The point mutation in the *dwf5* mutant allele is indicated with a line and the T-DNA insertion with a triangle. **b** Growth phenotypes of 4-week-old plants. Red, blue, and white arrowheads indicate young, mature, and old leaves (for definitions, see Fig. 1), respectively. Bar = 2 cm. **c** Representative images of LDs (OLE1-GFP, green) in leaves of 4-week-old *Col/OLE1* and *dwf5-8/OLE1* plants and LDs in wild-type leaves stained by Nile Red. The experiment was repeated 4 times with similar results. Bar = 50 μm. **d**, **e**, TLC analysis of neutral lipids. Lipids were extracted from leaves of 4-week-old plants and visualized with 5% sulfuric acid by charring (**d**). Relative contents of TAG were quantified by Image J (**e**). Data are mean ± SD of three biological replicates. Different letters indicate significant difference at *P* < 0.05, as determined by one-way ANOVA with Tukey's multiple comparisons test.

their immediate precursors without major changes in total free sterol content (Supplementary Table 1). The *dwf7-4* mutant is defective in the $\Delta^7$-sterol-C5-desaturase[32]. This mutant contains very limited amounts of all major sterols but accumulate an unusual sterol molecule $\Delta^7$- sitosterol and the total free sterol content is decreased slightly in *dwf7-4* compared with WT (Supplementary Table 1). The *cvp1-3 smt3-1* double mutant contains a fivefold higher level of campesterol, a 24-methyl-$\Delta^5$-sterol, while the amount of 24-ethyl-$\Delta^5$-sterols is reduced by 63% and total free sterol content remains largely unaltered (Supplementary Table 1). Introducing the *dwf1-2* into the *OLE1* line resulted in no significant change of TAG content in leaves. In contrast, disruption of DWF7 or SMT2/3 caused drastic decreases in TAG accumulation in the *OLE1* line (Fig. 3). Together, these results suggest that deficiency in 24-ethyl-$\Delta^5$-sterols and 24-ethylidene-$\Delta^5$-sterol is responsible for reduced TAG and LD accumulation in sterol mutants.

In addition to the changes in free sterol content and composition, all the sterol mutants tested, particularly *dwf5*, *dwf7*, and *cvp1-3 smt3-1* displayed significantly reduced levels of SEs (Supplementary Table 2). To test whether the deficiency in SEs affect TAG accumulation, we obtained T-DNA insertion mutants disrupted in PSAT1 and crossed the mutants with the *OLE1* line. As expected, disruption of PSAT1 caused major reductions in SE content in leaves (Supplementary Table 2). However, there were no significant changes in LD abundance and

TAG content in leaves and mature seeds of *psat1/OLE1* mutants compared with those of the *OLE1* line (Supplementary Fig. 5). Together, these results suggest that the decreased SE levels are not responsible for the LD-deficient phenotype in the sterol mutants.

**The *OLE1* transcript level correlates strongly with LD abundance.** To test whether sterols directly impact LD abundance via their effects on *OLE1* expression, we analyzed the *OLE1* transcript level in *dwf* mutants, two additional LD-deficient mutants (*M2-1* and *M3-1*) isolated from the same genetic screen as for *M1-7* and the phospholipid:diacylglycerol acyltransferase1 (PDAT1) knockout line carrying the *OLE1* transgene (*pdat1-2/OLE1*). Unlike sterol-deficient *dwf* mutants, *M2-1*, *M3-1*, and *pdat1-2/OLE1* showed normal growth phenotypes and thus presumably contained normal levels of sterols. However, like *dwf5* and *dwf7*, all these three LD-deficient mutants had drastic decreases in levels of the *OLE1* transcript and there is a strong positive correlation between the *OLE1* transcript levels and LD abundance in leaves (Supplementary Fig. 6). These results suggest that the expression level of the *OLE1* transgene is related to LD abundance rather than to amounts of sterols per se.

**TAG metabolism is not altered in *dwf5* and *dwf7* mutants lacking OLE1.** Mutants defective in DWF1, DWF4, DWF5, and DWF7 are all dwarf. Their morphology, growth, and

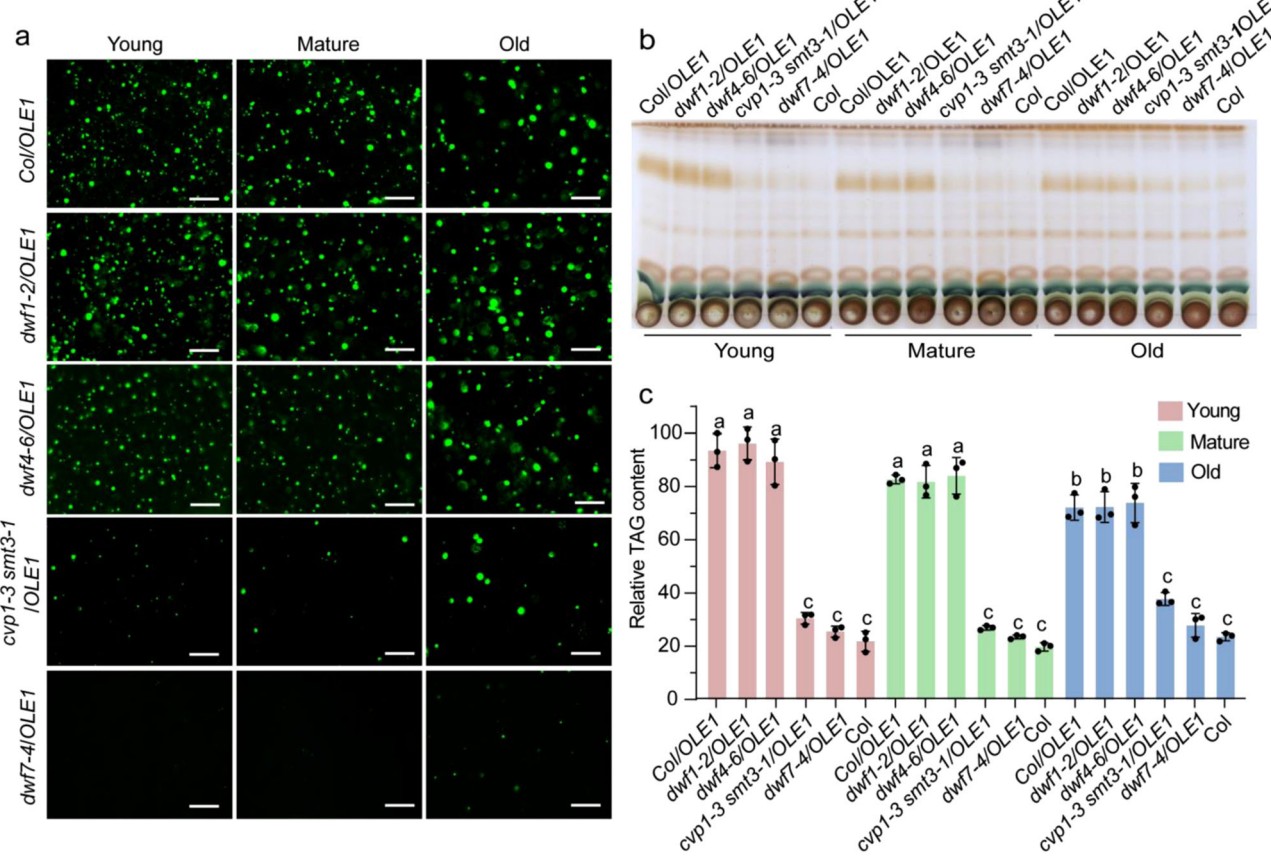

**Fig. 3 Disruption of DWF7 or SMT2/3 but not DWF1 and DWF4 results in defects in TAG and LD accumulation in leaves. a** Representative images of LDs (OLE1-GFP, green) in young, mature, and old (for definitions, see Fig. 1) leaves of 4-week-old plants. The experiment was repeated four times with similar results. Bar = 50 μm. **b, c** TLC analysis of neutral lipids. Lipids were isolated from leaves of 4-week-old plants and visualized with 5% sulfuric acid by charring (**b**). Relative contents of TAG were quantified by Image J (**c**). Data are mean ± SD of three biological replicates. Different letters indicate significant differences at *P* < 0.05, as determined by one-way ANOVA with Tukey's multiple comparisons test.

developmental patterns are similar among the mutants but quite different compared with WT (Supplementary Fig. 7). Therefore, to address the question of how the altered sterol profile affects TAG metabolism in the sterol mutants, we first compared rates of TAG synthesis and turnover among *dwf4-6*, *dwf5-8*, and *dwf7-4* mutants with or without the *OLE1* transgene in pulse-chase experiments with cold or radiolabeled oleic acid (18:1). TAG content in leaves of *dwf4-6*, *dwf5-8*, and *dwf7-4* was very low prior to feeding (Fig. 4a). Feeding with 0.5 mM 18:1 or 0.02 mM $^{14}$C-18:1 led to large increases in TAG content and radiolabeled TAG in all three *dwf* mutants. During the 24-h chase, TAG content and radiolabeled TAG decreased by 50% and more than 80%, respectively, in all the mutants lacking the *OLE1* transgene (Fig. 4a, b). The higher turnover rate of radiolabeled TAG than that of cold TAG may reflect less TAG accumulation in pulse-chase experiments with $^{14}$C-18:1 than with cold 18:1 since the concentration of the later is 25-fold higher than that of the former. Nevertheless, there were no significant differences in TAG levels either following the pulse or during the chase between *dwf4-6* and *dwf5-8* or *dwf7-4*, except that *dwf5-8* appeared to contain slightly higher amounts of radiolabeled TAG following the pulse with $^{14}$C-18:1. These results suggest that disruption of DWF5 or DWF7 has no major impact on TAG metabolism in leaves.

To further investigate lipid metabolism in the sterol mutants, we carried out $^{14}$C-acetate pulse-chase experiments using detached leaves. The rates of labeled acetate incorporation into total fatty acids, as well as into TAG were similar between *dwf1/4*

and *dwf5/7* (Supplementary Fig. 8a, b). Arabidopsis PDAT1 plays a critical role in TAG synthesis in leaves[31]. Assays for PDAT activity in microsomal membranes revealed that sterols had no effect on TAG formation from $^{14}$C-labeled phosphatidylcholine (PC) in leaves (Supplementary Fig. 8d). During the chase, the label in TAG decreased in all the sterol mutants with no significant differences between *dwf1/4* and *dwf5/7*, reflecting a similar TAG degradation rate in these sterol mutants (Supplementary Fig. 8c). Together, these results suggest that deficiency in 24-ethyl-Δ$^5$-sterols and 24-ethylidene-Δ5-sterol in the mutants lacking OLE1 has no effects on rates of fatty acid synthesis, TAG synthesis, and turnover.

**Leaf TAG turnover is accelerated in *dwf5/OLE1* and *dwf7/OLE1*.** In the presence of the *OLE1* transgene, *dwf4-6* leaves contained a much higher amount of TAG than leaves of *dwf5-8* or *dwf7-4* following the incubation with 0.5 mM 18:1, likely due to a higher amount of preexisting TAG accumulation in *dwf4-6/OLE1* than in *dwf5-8/OLE1* and *dwf7-4/OLE1* (Fig. 4a). In line with this possibility, the amount of radiolabeled TAG in *dwf4-6/OLE1* following the pulse with $^{14}$C-18:1 was similar to that in *dwf5-8/OLE1* and *dwf7-4/OLE1*, suggesting again that disruption of DWF5 or DWF7 has no impact on the rate of TAG synthesis (Fig. 4b). During the 24-h chase, TAG levels decreased slightly in *dwf1-2/OLE1* and *dwf4-6/OLE1* but markedly in *dwf5-8/OLE1* and *dwf7-4/OLE1* (Fig. 4a, b, Supplementary Fig. 9). Like the situation in *dwf* mutants lacking OLE1, radiolabeled TAG accumulated following $^{14}$C-18:1 feeding decreased faster than TAG

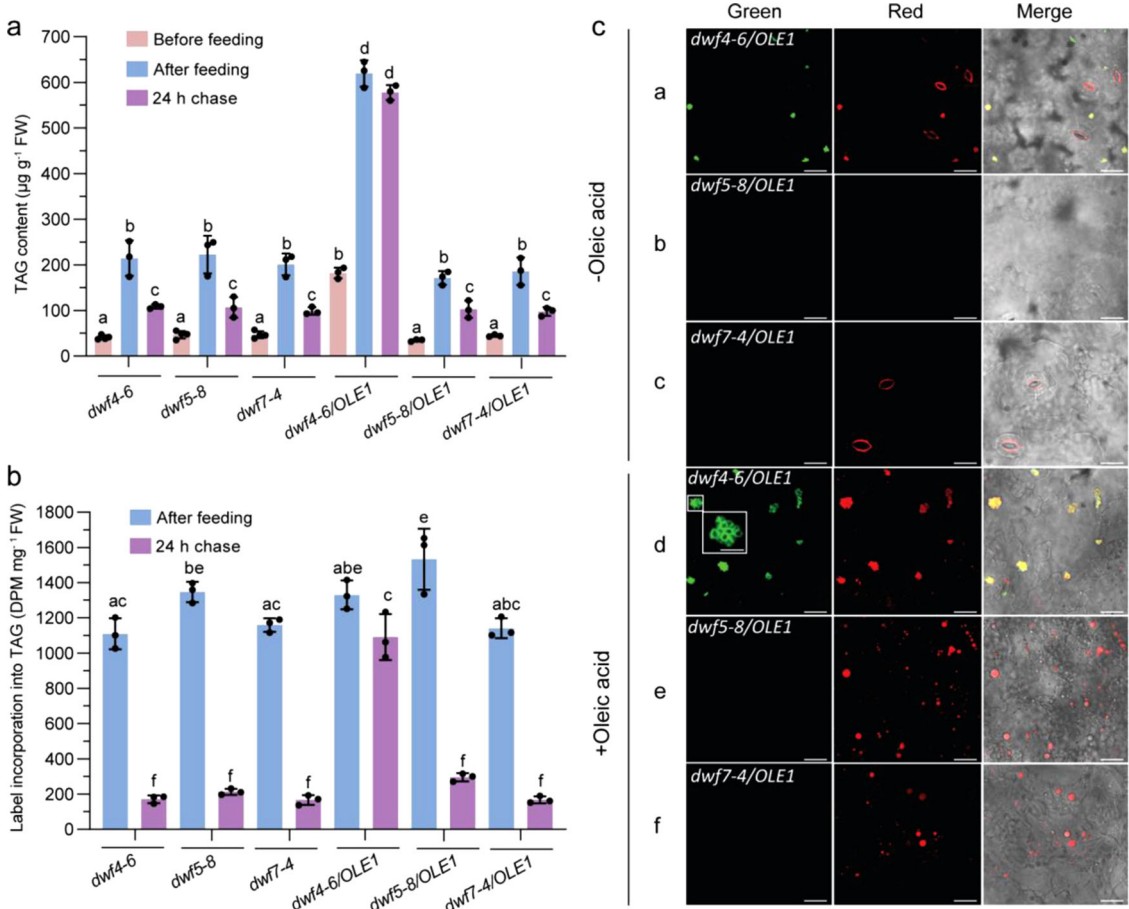

**Fig. 4 TAG metabolism and LD accumulation in leaves of *dwf* mutants as revealed by oleic acid feeding. a** TAG contents in leaves before oleic acid feeding, after 0.5 mM cold oleic acid feeding for overnight and after 24 h of chase. Lipids were extracted from leaves and quantified by GC-MS. Four biological replicates were performed for *dwf4-6*, dwf5-8, and *dwf7-4* before feeding while the others had three biological replicates. **b** Label incorporation into TAG in leaves after $^{14}$C-oleic acid feeding and after 24 h of chase. Lipids were extracted from leaves and separated by TLC. Radiolabeled TAGs were quantified by liquid scintillation counting. Three biological replicates were carried out for each sample. **c** LDs in leaves after 0.5 mM cold oleic acid feeding for overnight. LDs were stained by BODIPY red and observed with a confocal scanning laser microscope. The experiment was repeated 3 times with similar results. Bars = 20 μm (5 μm in insert). In **a** and **b**, data are mean ± SD. Different letters indicate significant differences at $P < 0.05$, as determined by one-way ANOVA with Tukey's multiple comparisons test.

accumulated following feeding with cold 18:1 in all *dwf/OLE1* mutants, particularly in *dwf5-8/OLE1* and *dwf7-4/OLE1*, during the chase. Together, these results suggest that disruption of DWF5 or DWF7 accelerates TAG turnover in the *OLE1* transgenic line.

**Leaves of *dwf5/OLE1* and *dwf7/OLE* mutants accumulate large-sized individual LDs lacking oleosin upon oleic acid feeding.** Overexpression of GFP-tagged OLE1 in Arabidopsis leaves resulted in the clustering of small LDs[31]. These OLE1-GFP-decorated LD clusters were clearly seen in leaves of *dwf4-6/OLE1* and were colocalized with LD clusters stained with a neutral lipid dye, BODIPY red, as expected (Fig. 4c, panel a). In contrast, both the green and red fluorescent signals were barely detectable in *dwf5-8/OLE1* and *dwf7-4/OLE1* leaves (Fig. 4c, panel b and c). To test whether sterol deficiency affects LD organization and morphology, detached leaves of *dwf4-6/OLE1*, *dwf5-8/OLE1*, and *dwf7-4/OLE1* were first incubated with 0.5 mM 18:1 overnight and then stained with BODIPY red. Consistent with increased TAG content, feeding with 18:1 resulted in an increase in the size and number of OLE1-GFP-decorated LD clusters in *dwf4-6/OLE1* and the vast majority of green fluorescent signals colocalized with BODIPY-stained LD clusters. The individual LD in the clusters

was small with diameters of less than 1 μm (Fig. 4c, panel d). Strikingly, even though 18:1 feeding led to the accumulation of large amounts of TAG (Fig. 4a), green fluorescent signals were barely detected in 18:1-fed *dwf5-8/OLE1* and *dwf7-4/OLE1* leaves (Fig. 4c, panel e and f). BODIPY staining, on the other hand, revealed the presence of abundant large-sized LDs with diameters of up to 10 μm, and most of them were present in solitary forms (Fig. 4c, panel e and f).

**Mutants disrupted in DWF5 and DWF7 have reduced seed TAG content and altered seed morphology.** The data presented thus far suggest that deficiency in 24-ethyl-Δ5-sterols and 24-ethylidene-Δ5-sterol results in the formation of unstable LDs lacking OLE1 coating even in the presence of the constitutively expressed *OLE1* transgene. The endogenous oleosin genes are not expressed in leaves but in developing seeds during seed oil accumulation. To exclude the possibility that the observed LD phenotypes in leaves of sterol mutants are due to artifacts arising from the ectopic expression of OLE1 or oleic acid feeding, we sought to investigate the role of sterols in oil accumulation in seeds of sterol mutants with or without the *OLE1* transgene. To this end, we first analyzed sterol content and composition in seeds. Total free sterol levels in mature seeds were increased in

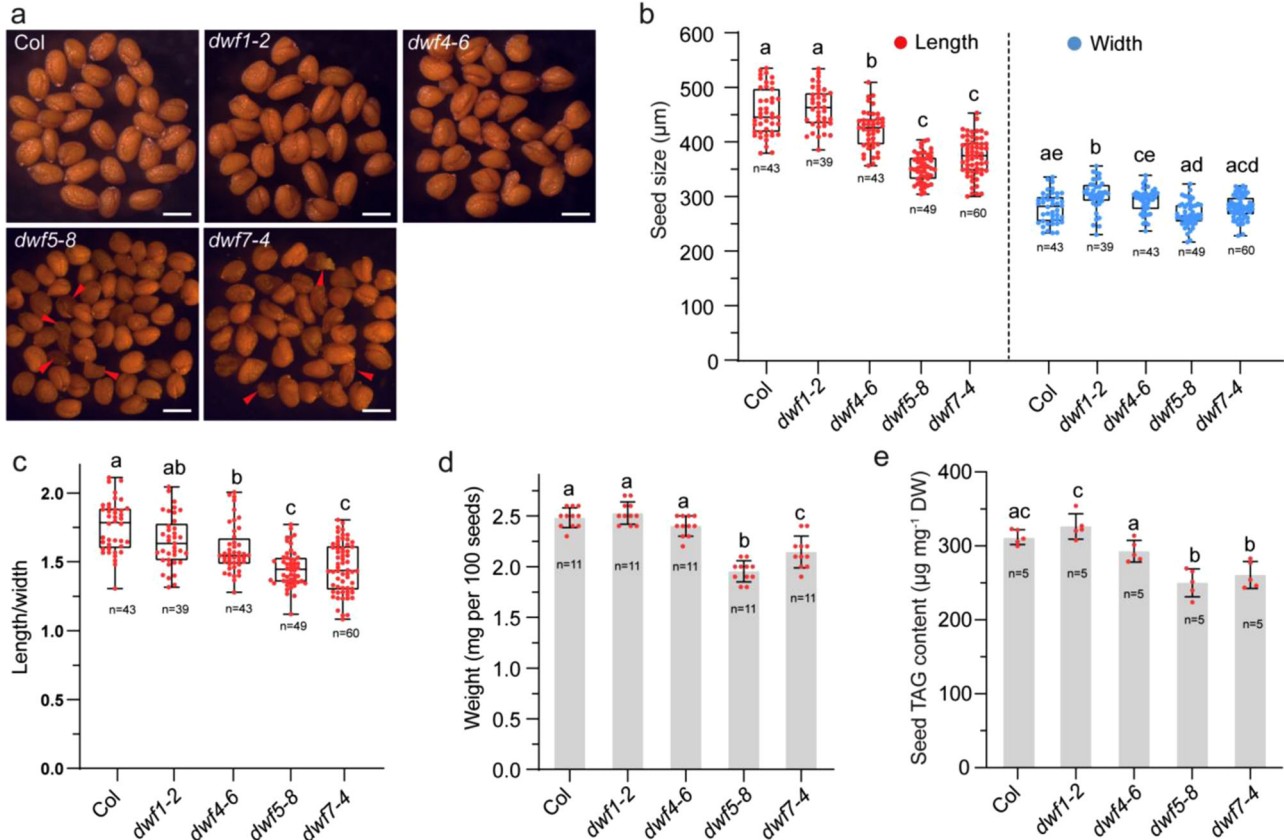

**Fig. 5 Reduced seed TAG content and altered seed morphology in *dwf5* and *dwf7* mutants. a** Representative phenotypes of dry seeds of wild type (Col), *dwf1-2*, *dwf4-6*, *dwf5-8*, *dwf7-4*. Red triangles indicate severe wrinkled seeds. Bar = 400 μm. **b** Seed size. **c** Length to width ratios of seeds. Numbers in **b** and **c** represent the number of seeds analyzed for data collection. **d** Weight per 100 dry seeds. **e** Seed TAG contents quantified by GC-MS. In **b** and **c**, the box plot boundaries reflect the interquartile range, the center line is the median and the whiskers represent 1.5 × the interquartile range from the lower and upper quartiles. Numbers in **b** and **c** represent the number of seeds analyzed for data collection. Data in **d** and **e**, *n* represent the number of biological replicates. Data are mean ± SD. In **b**, **c**, **d**, and **e**, different letters indicate significant differences at *P* < 0.05, as determined by one-way ANOVA with Tukey's multiple comparisons test.

*psat1* mutants, remained largely unchanged in *cvp1-3 smt3-1* and were decreased by 72, 75, and 56% in *dwf5-8*, *dwf5-10/OLE1*, and *dwf7-4*, respectively, compared with the corresponding controls (Supplementary Table 3). Like what was found in leaves of *dwf5* and *dwf7* mutants (Supplementary Table 1), the decreases in total free sterols in mutant seeds were largely attributable to decreases in levels of 24-ethyl-$\Delta^5$-sterols including sitosterol and stigmasterol, whereas unusual sterols such as stigmasta-5,7,22-trienol and stigmasta-5,7-dienol in *dwf5-8* and $\Delta^7$-sitosterol in *dwf7-4* were increased (Supplementary Table 3).

We next examined the morphology of mature seeds of *dwf* mutants. Both *dwf5-8* and *dwf7-4* seeds were smaller and rounder than WT seeds (Fig. 5a) and their length, width, and length/width ratios were decreased compared with WT and *dwf1-2* (Fig. 5b, c). In addition, seed weights of the *dwf5-8* and *dwf7-4* mutants were significantly decreased compared to the weight of WT seeds (Fig. 5d). Unlike the mutants upstream of 24-methylenelophenol, such as *smt1* and *hyd1*, which commonly show defects in embryogenesis, downstream mutants *cvp1/smt2*, *dwf7*, *dwf5*, and *dwf1/dwf1-2* do not exhibit altered embryogenesis[32]. However, like what was found in *dwf5-1* seeds[33], a substantial proportion of seeds in *dwf5-8* and *dwf7-4* were dark brown in color and wrinkled to some extent (Fig. 5a). A possible reason of the seed phenotype is TAG reduction in these wrinkled seeds. Indeed, TAG content per milligram seeds decreased significantly by 17.6% and 11.6% in *dwf5-8* and *dwf7-4* seeds, respectively, compared with WT, whereas no significant change was found in

*dwf1-2* and *dwf4-6* seeds (Fig. 5e). Seeds of *dwf5-10/OLE1* showed a much more severe wrinkled appearance than that of *dwf5-8* mutant, with about 37.4 ± 2.4% seeds being affected in the former (Supplementary Fig. 10a–d). Consequently, TAG content in *dwf5-10/OLE1* decreased by up to 40% compared with *Col/OLE1* seeds (Supplementary Fig. 10e). In addition, oil levels in developing embryos decreased by 73 and 57% at the bent cotyledon stage in *dwf5-8* and *dwf7-4*, respectively and by 50 and 20% at the mature green stage in *dwf5-8* and *dwf7-4*, respectively (Supplementary Fig. 11).

**Disruption of DWF5 or DWF7 results in an increase in LD size in seeds**. To investigate the effect of sterol deficiency on LD morphology, we carried out the microscopic analysis of LDs in embryos from mature seeds. Transmission electron microscopy (TEM) imaging revealed that LDs in *dwf1-2* seed cells showed a similar size distribution pattern to that of the WT. However, LDs in *dwf5-8* mutants were more heterogeneous, larger, and rounder compared with LDs in WT and *dwf1-2* (Fig. 6a). Quantitative analysis revealed that both the length of long axis and short axis of LDs in the *dwf5-8* mutant were significantly increased while the ratio of the length of long axis to short axis was significantly decreased compared with LDs in WT seeds (Fig. 6b, c). Consequently, the LD number per microscopical field was significantly decreased in *dwf5-8* seeds (Fig. 6d). LDs in *dwf7-4* seeds were also enlarged compared with those in WT

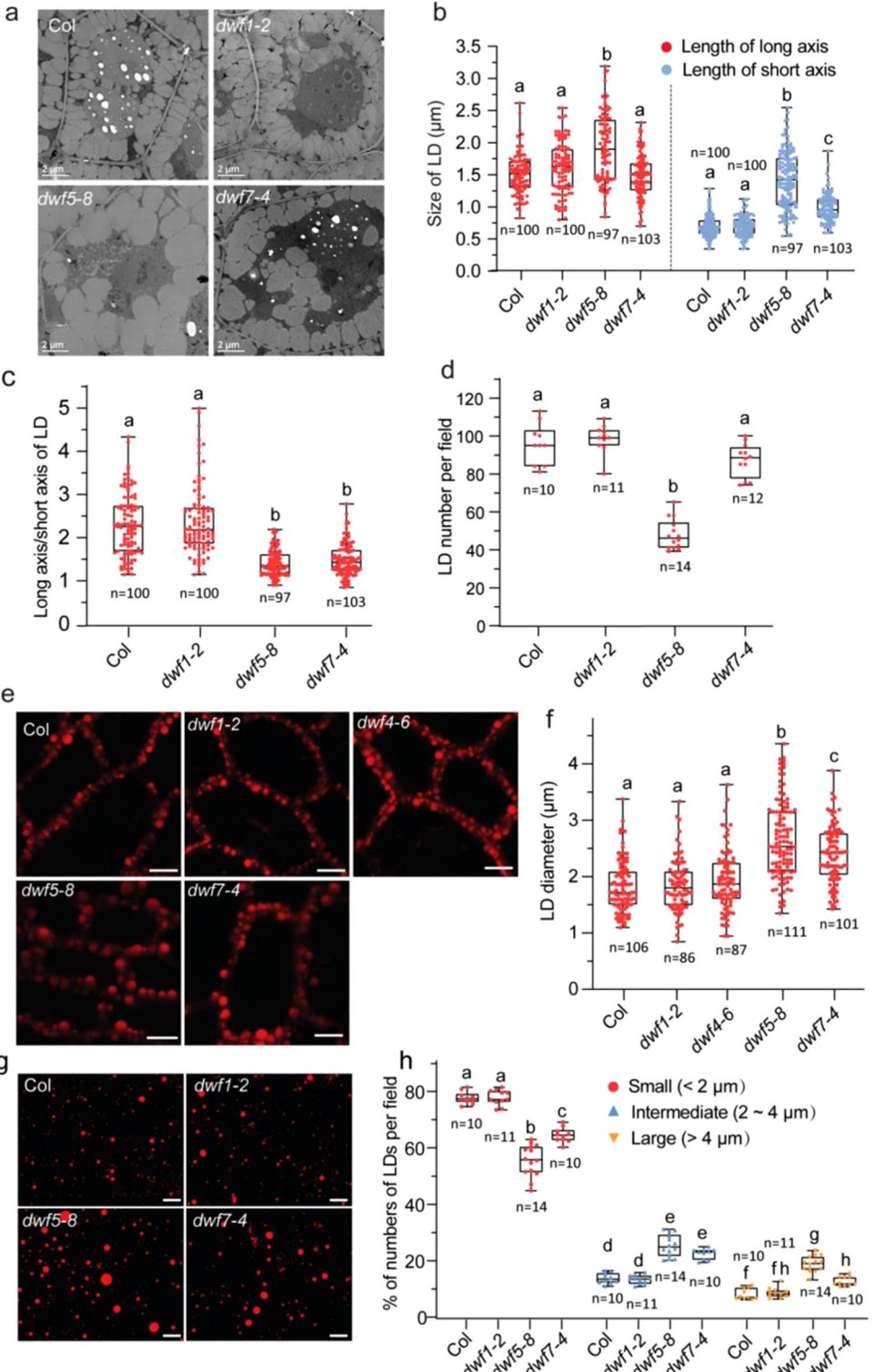

seeds, with increased length of short axis but the comparable length of long axis (Fig. 6a–d).

To examine changes in LD morphology in vivo, 3-day-old seedlings were stained with Nile Red and observed under confocal microscopy. Again, we found that LD size was significantly enlarged in *dwf5-8*, *dwf7-4* hypocotyl cells than that of WT (Fig. 6e, f). To obtain more quantitative information on changes in LD size, LDs from mature seeds were isolated by floating centrifugation, and numbers of purified LDs were counted and grouped into three different sizes (Fig. 6g, h). The results revealed

significant decreases in the number of small LDs in *dwf5-8* and *dwf7-4* mutants and concomitant increases in the number of intermediate- and large-sized LDs. In addition, there was a significant increase in the number of large-sized LDs at the expense of small-sized ones in *dwf5-8* compared with *dwf7-4*, mirroring more severer decreases in amounts of 24-ethyl-$\Delta^5$-sterols in the former than in the later (Supplementary Table 3). No significant difference in the numbers of LDs in all three size categories was found between WT and *dwf1-2* (Fig. 6g, h). In summary, microscopic analysis of LDs in *dwf5* and *dwf7* seeds

**Fig. 6 Changes in LD morphology in *dwf5* and *dwf7* mutants. a** TEM images of LDs in mature seeds of wild type and *dwf* mutants. The experiment was repeated 3 times with similar results. Bar = 2 μm. **b** Size of LDs. **c** Long axis to short axis ratios of LDs. **d** LD number per field of TEM. **e** Representative confocal images of LD in the hypocotyl epidermis of wild type and *dwf* mutants. LDs of 3-day-old seedlings were stained with Nile Red and visualized in the hypocotyl epidermis next to the radicle by confocal microscope. The experiment was repeated three times with similar results. Bar = 10 μm. **f** Diameter distribution of LDs from hypocotyl epidermis of wild type and *dwf* mutants. **g** Representative images of isolated LDs. LDs were isolated from dry seeds, stained with Nile Red and visualized with a Zeiss epifluorescence microscope. The experiment was repeated 3 times with similar results. Bar = 20 μm. **h** Percentages of different sizes of isolated LDs per microscopic field. Small LDs, diameters < 2 μm; intermediate LDs, diameters between 2 and 4 μm; large LDs, diameters larger than 4 μm. Numbers in **d** and **h** represent the number of microscopic fields analyzed for data collection. Numbers in **b**, **c** and **f** indicate the number of LDs analyzed. In **b**, **c**, **d**, **f** and **h**, the box plot boundaries reflect the interquartile range, the center line is the median and the whiskers represent 1.5 × the interquartile range from the lower and upper quartiles. Different letters indicate significant differences at $P < 0.05$, as determined by one-way ANOVA with Tukey's multiple comparisons test.

reinforces the notion that sterols are required for proper LD formation.

**Seed oleosin protein abundance is markedly decreased in *dwf5* and *dwf7* mutants.** The increased LD size in seeds of *dwf5* and *dwf7* mutants prompted us to investigate the impact of sterol deficiency on seed oleosin protein levels, since LD size in seeds is known to be determined largely by oleosin protein abundance[26]. For this purpose, we first examined OLE1-GFP signals in developing embryos of sterol mutants overexpressing *OLE1*. Although there is evidence that the 35 S CaMV promoter is not very active in developing seeds[34], abundant OLE1-GFP-coated LDs were observed in developing embryos of *Col/OLE1* and *dwf1-2/OLE1* (Fig. 7a). Disruption of DWF5 or DWF7 led to marked decreases in OLE1-GFP-coated LDs in developing embryos of the *OLE1* line, and again *dwf5-8/OLE1* showed a more severe phenotype than *dwf7-4/OLE1*. Western blot analysis confirmed drastic deceases in OLE1-GFP levels in developing embryos of *dwf5-8/OLE1* and, to a lesser extent, in embryos of *dwf7-4/OLE1*, compared with the *Col/OLE1* and *dwf1-2/OLE1* (Fig. 7b, c). Similarly, the abundance of OLE1-GFP-coated LDs was also found to markedly decrease in cotyledon and root of 6-day-old seedlings of *dwf5-8/OLE1* and *dwf7-4/OLE1* compared with *Col/OLE1* and *dwf1-2/OLE1* (Supplementary Fig. 12).

Oleosins account for up to 79% of total LD proteins in seeds of Arabidopsis[35] and all five oleosin isoforms can be easily resolved by sodium dodecyl sulfate-polyacrylamide gel electrophoresis (SDS-PAGE)[27]. To gain more quantitative information on the effect of sterol deficiency on oleosin levels, LDs were purified from mature seeds of WT, *dwf1-2*, *dwf5-8*, and *dwf7-4*, and LD-associated proteins were loaded on a same amount of TAG basis and separated by SDS-PAGE. As expected, the isolated LD fractions were particularly enriched with oleosin proteins migrating within the 10-20 kDa size range (Fig. 7d). Among them, OLE1 was the most abundant isoform, followed by OLE2, OLE3, and OLE4, whereas OLE5, which has the smallest molecular mass[27], was not detected in our experiments. Compared with WT, both the *dwf5-8* and *dwf7-4* showed marked decreases in levels of all four oleosin proteins. Results of image analysis showed that the total amounts of OLE1, OLE2, and OLE3 decreased by 50 and 35% in *dwf5-8* and *dwf7-4*, respectively, compared with the WT, whereas no differences were found between *dwf1-2* and WT (Fig. 7e–g). On the other hand, the level of a protein band migrating at 30 kDa (mostly likely the LD protein caleosin based on its molecular weight) remained largely unaltered in *dwf5-8* and *dwf7-4* mutants compared with the WT (Fig. 7h).

## Discussion

The lipid and protein factors involved in regulating LD biogenesis and growth remain unknown, particularly in plants. In this study, a comprehensive genetic, biochemical, and cell biological analysis uncovers a BR-independent role of sterols in LD accumulation in plants. Sterols can function as signaling molecules in regulating gene transcription in plants[6]. Our analysis, however, revealed a strong positive correlation between the *OLE1* transcript level and LD abundance in *dwf* mutants and other LD-deficient mutants with normal growth phenotypes. Similarly, TAG and LD accumulation has often been found to concur with increases in oleosin protein and transcript levels in developing seeds. Disruption of WRINKLED1, a key transcription factor controlling the expression of genes involved in the synthesis of fatty acids but not oleosins, results in drastic decreases in levels of both the TAGs and oleosin transcripts in seeds[36]. In addition, TAG content and oleosin expression have been found to decrease concomitantly in seeds of a mutant disrupted in TAG synthesis[37]. Together, these results strongly suggest that oleosin transcript levels are controlled by LD abundance and imply that sterols impact oleosin gene expression indirectly via their effects on LD assembly. Sterols may be involved in the regulation of RNA processing, stability, and translation, but to the best of our knowledge, no such regulatory roles have been reported in any model systems. On the other hand, sterols as key regulators of the physical properties and microdomain organization of cellular membranes are well documented. Therefore, A more plausible explanation for the phenotype is that sterol deficiency or imbalanced sterol composition disrupt normal membrane properties and domain organization necessary for membrane-associated events during the coordinated assembly of LDs from oil and oleosins.

Oleosins function not only as the key determinants of LD stability and size but also in the early stages of LD formation such as LD budding from the ER into cytosol[24]. Arabidopsis seed LDs contain five oleosin proteins, with OLE1, OLE2, and OLE3 as the major isoforms (Fig. 7d). A previous study showed that combined knockout of OLE1, OLE2, and OLE4 genes only results in an about 10% decrease in seed oil[27]. In *dwf5-8* mutant seeds, a 50% decrease in amounts of OLE1, OLE2, and OLE3 was accompanied by up to 40% decreases in oil content in mature seeds (Fig. 5e and Supplementary Fig. 10). These results suggest that the decreased oil content in *dwf5* seeds cannot be solely explained by changes in oleosin levels. Based on LD size distribution shown in Fig. 6h, we estimated that the increased LD size in *dwf5* only resulted in a small decrease (less than 5% compared with WT) in LD surface area, suggesting that the decreased oleosin levels are not due to a decrease in LD surface area.

Due to their long hydrophobic stretch, oleosins are predicted to be unstable when present in ER membranes and become stable when associated with LDs[25]. Likely for this reason, the transcript levels of genes involved in oil and oleosin synthesis are coordinately regulated during seed development[38]. In addition, oleosins are synthesized in ER domains alongside with the synthesis of TAGs[39] and the assembly of nascent LDs[40]. Presumably, this

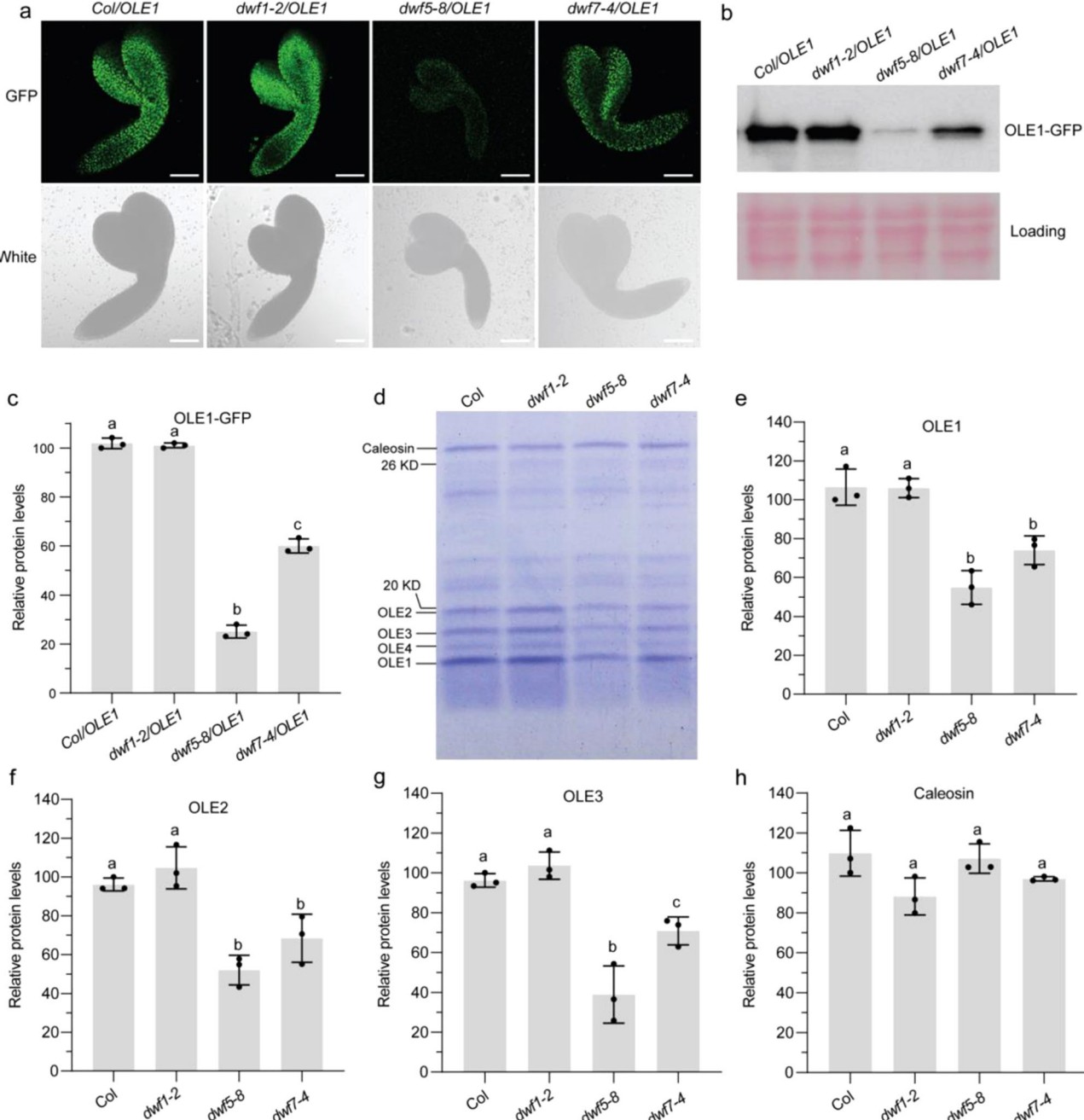

**Fig. 7 Seed oleosin protein abundance is decreased in *dwf5* and *dwf7* mutants. a** Representative images of LDs (OLE1-GFP, green) in embryos at the bent cotyledon stage. The experiment was repeated three times with similar results. Bar = 100 μm. **b** Western blotting with anti-GFP antibody showing OLE1-GFP protein levels in embryos of *Col/OLE1*, *dwf1-2/OLE1*, *dwf5-8/OLE1*, and *dwf7-4/OLE1* at the bent cotyledon stage. Equal loading of proteins are shown by Ponceau S staining. **c** Relative OLE1-GFP protein levels quantified using Image J software. **d–h**, Oleosin and caleosin protein levels in seed LDs. LDs were purified from dry seeds. Proteins of LDs equivalent to 15 μg of TAGs were subjected to SDS-PAGE with Coomassie blue staining (**d**). Relative oleosin and caleosin protein levels were quantified using ImageQuant TL software (**e–h**). In **c**, **e**, **f**, **g**, and **h**, data are mean ± SD. of three biological replicates. Different letters indicate significant differences at *P* < 0.05, as determined by one-way ANOVA with Tukey's multiple comparisons test.

spatiotemporal regulation enables a concomitant accumulation of TAG and oleosins and efficient transfer of oleosins to LDs[25,39,40] during seed filling. Analysis of temporal changes in oil accumulation in developing seeds showed that the decrease in TAG content was much more pronounced in the bent cotyledon than in the green mature stage of developing embryos in both the *dwf5* and *dwf7* mutants (Supplementary Fig. 11). During early embryo development, *dwf5* and *dwf7* mutants contained 57–73% lower levels of TAG than *dwf1*. If oleosin synthesis in *dwf5* and *dwf7*

mutants continued at a similar rate to *dwf1*, most of these nascent proteins would be expected to remain in the ER and be degraded quickly. During the transition from the bent cotyledon to the green mature stage, the rates of TAG accumulation were threefold higher in *dwf5* and *dwf7* than the rate found in *dwf1*. Such high rates of TAG accumulation may not be matched by a similar rate of oleosin synthesis. Together, these results suggest that disruption of the coordinated synthesis of TAG and oleosins and their deposition into LDs may be one of the major factors contributing

to the formation of LDs with the less oleosin coat in mutants deficient in sterols.

Given so much is still unknown regarding the mechanism of LD biogenesis, we can only speculate on the functions of sterols in neutral lipid storage in plants. Like DAG, sterols are non-bilayer, conical lipids with a small polar head group and a large hydrophobic body. When present in the biological membranes, sterols induce lipid packing defects, increase membrane surface tension and alter other aspects of membrane properties in a similar way as DAG does[41,42]. In addition, the presence of sterols has been shown to greatly amplify the membrane perturbing effects of DAG[43]. DAG has been suggested as one of the key lipids in regulating membrane biophysical properties and hence initial oil lens formation as well as LD budding[18,44,45]. Therefore, it is possible that sterols, either acting alone or synergistically with DAG, are involved in regulating membrane properties necessary for LD formation. Indeed, a recent study has shown that cholesterol promotes LD initiation and the packaging of TAGs into LDs[45]. In addition to its roles in the early steps of LD biogenesis, DAG has been shown to be important for protein targeting to LDs[46], likely via its ability to induce local phospholipid packing defects on LD surface which function as membrane docking sites for some of LD-associated proteins[44]. Free sterols are known to be amply present in the LD phospholipid monolayer[47,48]. Therefore, the second possible scenario is that deficiency in sterols impairs the ability of LD monolayer surface lipids to induce lipid packing defects and hence the recruitment of LD-associated proteins necessary for LD initiation, budding, growth, and protection. In line with this, sterols have been shown to affect the physical properties of the LD surface and hence the targeting of TAG-synthesizing enzymes to LDs in mammals[49]. The third possibility is that deficiency in sterols impairs the ability of ER membranes to generate sterol-rich microdomains necessary for LD biogenesis. In support, there is ample evidence that LDs are formed at discrete sites of the ER[17,18,44]. In addition, sterol-rich ER domains have been implicated in LD biogenesis[10,11] and the synthesis of neutral and membrane lipids[9]. In mammalian cells, disruption of lipid microdomains enriched in cholesterol inhibits LD formation[11].

The most important structural determinants of sterols in modulating membrane properties and domain organization are the presence of a side chain of 8 to 10 carbon length at position C-17, a hydroxyl group at position C-3, and a $\Delta^5$ double bond between carbon atoms C-5 and C-6 in the tetracyclic ring[50-52]. In particular, it has been shown that the $\Delta^5$ double bond is required for the preferential interaction of sterols with saturated phospholipids and this interaction serves as a driving force for the formation of membrane microdomains[53-55]. Importantly, a shift of the double bond position or introduction of a second double bond in the tetracyclic ring changes the membrane domain forming properties of the sterols[56,57] and renders sterols far less effective in mediating the sterol-phospholipid interaction[50]. The importance of the $\Delta^5$-sterols in regulating membrane properties and hence LD biogenesis is clearly illustrated in dwf5 and dwf7 mutants, which accumulate unusual sterols at the expense of $\Delta^5$-sterols and are severely defective in LD formation. On the other hand, disruption of DWF1 causes drastic decreases in sitosterol and campesterol with concomitant increases in isofucosterol and 24-methylenecholesterol but no significant effects on total $\Delta^5$-sterol content and LD formation. These results suggest that isofucosterol and 24-methylenecholesterol are functionally interchangeable with sitosterol and campesterol, respectively in LD biogenesis in plants. In fact, isofucosterol and 24-methylenecholesterol differ structurally from sitosterol and campesterol only by a single double bond at C-24 of the side chains and the key structural features required for optimal interaction with phospholipids are present in both types of sterols.

Among the major end products of plant sterols, campesterol exhibits the strongest ability to promote membrane domain formation and to organize sterol spatial distribution in membranes[57]. Therefore, a drastic increase in campesterol with a concomitant decrease in sitosterol may disrupt the balanced ratio of campesterol to sitosterol thus the optimal membrane environments necessary for LD formation and thus contribute to LD deficiency phenotype in cvp1-3 smt3-1. In addition to disturbed sterol composition, changes in overall sterol content appear also to impact LD biogenesis in plants. This notion is supported by several lines of evidence. First, total free sterol levels are reduced only in dwf5 mutants severely deficient in LD biogenies. Second, among all the sterol mutants analyzed, dwf5 has the lowest level of free sterols and this mutant exhibits the most severe LD deficiency phenotype in both leaves and seeds. Third, leaf senescence is known to be associated with an increased accumulation of sterols[13], TAG levels, LD number (Figs. 1–3) and total sterol content (Supplementary Fig. 13) are all increased in old leaves of both dwf5 and dwf7. Fourth, dwf5 and dwf7 mutants contained higher levels of 24-ethyl-$\Delta^5$-sterols in seeds (Supplementary Table 3) than in leaves (Supplementary Table 1) and both mutants showed less severe defects in storage lipid accumulation in seeds than in leaves. Since most of the seed mass is storage lipids and proteins, it seems likely that actual levels of free sterols in cellular membranes of dwf5 and dwf7 mutants are much higher than those reported in Supplementary Table 3. Alternatively, the observed more pronounced defects in LD accumulation in leaves than in seeds may reflect a different mechanistic basis underlying LD biogenesis between these two organs.

In summary, this study reveals a previously unrecognized role of sterols in LD biogenesis in plants. We propose that sterols regulate LD formation by controlling physicochemical properties of membranes or ER membrane domain organization. Further studies are required to dissect the exact mechanism by which changes in sterol composition and content affect LD formation in plants.

## Methods

**Plant materials and growth conditions**. *Arabidopsis thaliana* Columbia-2 ecotype was used for all experiments described. The mutants dwf1-2(CS8100)[58], dwf5-8 (SALK_127066)[33], bri1-3[59], cvp1-3smt3-1[60], pdat1-2[31], psat1-1 (SALK_037289), and psat1-2 (SALK_117091)[13] were described previously. The dwf4-6 (CS449028) and dwf7-4 (CS442813) mutants were obtained from the Arabidopsis Biological Resource Centre (ABRC) and verified by PCR. The *OLE1-GFP* overexpressing line was described in Fan et al.[31].

Surface-sterilized seeds were imbibed for 2 days at 4 °C in dark and then germinated on half-strength Murashige and Skoog (MS) medium with 0.6% (w/v) agar and 1% (w/v) sucrose in an incubator (22/18 °C day/night temperature; 16-h-light/8-h-dark photoperiod; and 80–120 µmol m$^{-2}$ s$^{-1}$ light intensity). Ten-day-old seedlings were transferred to soil and grown in a climate-controlled growth chamber (22/18 °C day/night temperature; 16-h-light/8-h-dark photoperiod; and 100–150 µmol m$^{-2}$ s$^{-1}$ light intensity).

**Isolation of M1-7, map-based cloning, and whole-genome sequencing**. Seeds of the transgenic plants overexpressing OLE1 fused with GFP were mutagenized with ethyl methanesulfonate (EMS) and leaves of 3-week-old M$_2$ plants were used to screen LD mutants using an epifluorescence microscope (Carl Zeiss; Axiovert 200 M). One dwarf mutant named M1-7 with only very few LDs was isolated. For map-based cloning, the M1-7 mutant was crossed with the wild-type plants of Landsberg erecta ecotype. About 60 F2 plants with M1-7 mutant phenotype were used for rough mapping. Mapping markers (Supplementary Table 4) were designed according to Arabidopsis mapping platform (AMP) website (http://amp.genomics.org.cn/)[61].

For the whole-genome sequencing, young leaf tissue from 102 F$_2$ lines of the mapping population were pooled, genomic DNA was extracted using DNeasy plant Mini Kit (QIAGEN, cat# 69104) and the genomic library preparation and sequencing were done in Weill Cornell Medical College Genomics Resources Center. The DNA was sequenced using a HiSeq2500 (Illumina) with 100-bp single-end reads. The clean reads were aligned to the WT reference genome and reliable

SNPs in the region of chromosome 1, where the mutation of *M1-7* was delimited by rough mapping, were identified.

**Lipid and fatty acid analyses**. For leaf TAG quantification, neutral lipids were extracted from leaves with chloroform/methanol/formic acid (1:2:0.1, by volume) and separated on silica plates (Silica Gel 60, EMD Millipore Corporation) by thin-layer chromatography (TLC) using a solvent system consisting of hexane/diethyl ether/acetic acid (70:30:1, by volume). TAG on TLC plates was visualized by spraying 5% $H_2SO_4$ followed by charring and the relative TAG band intensity was measured using the ImageJ software. Seeds harvested from plants grown side-by-side in the same flat were used for seed TAG quantification. In brief, 15 dry seeds or geminating seeds were incubated in 1 ml of 2.5% sulfuric acid in methanol (v/v) at 90 °C for 2 h. Five microgram C17:0 was added as an internal standard prior to transmethylation. Fatty acyl methyl esters were extracted with hexane and quantified by an HP5975 gas chromatography–mass spectrometer (Hewlett–Packard) fitted with 60 m × 250-μm SP-2340 capillary column (Supelco)[31].

**Microscopy**. Fresh leaf tissue was mounted on slides with water and observed under a Zeiss epifluorescence microscope (Carl Zeiss; Axiovert 200 M) equipped with a GFP filter set 38 (Carl Zeiss). For LD imaging in hypocotyl, 3-day-old seedlings germinated on ½ MS medium without sucrose were stained with 10 μg/ml Nile Red (Sigma-Aldrich) and examined using a Leica TCS SP5 laser scanning confocal microscope with excitation at 543 nm and emission at 570–700 nm. For leaf LD imaging in oleic acid feeding experiments, leaves were washed with water for four times with shake (5 min each) after feeding, blotted dry, then stained with 10 μg/ml BODIPY 558/568 C12 (Invitrogen, cat# D3835) and examined by confocal microscopy. For transmission electron microscopy (TEM), mature green embryos of the same developmental stage at about 13 to14 days after pollination were collected, and the Ultrabed Kit (Electron Microscopy Sciences) were used for infiltration and embedding. Samples were then sectioned and stained with 2% uranyl acetate and lead citrate before viewing under a JEM-1400 LaB6 120-keV transmission electron microscope (JEOL, USA). The size of LDs was measured using the ImageJ software.

**Seed and LD size measurements**. Seeds from different genotype were harvested at the same time from plants grown side-by-side in the same flat. Seeds pictures were taken with Leica M125 dissecting microscope with Jenoptik ProgRes SpeedXT Core 5 CCD microscope camera. The length and width of the seeds were measured using the ImageJ software.

**Sterol measurements by GC-MS**. Free sterols were extracted as described previously[62] with some modifications. Briefly, about 2 g mature leaf of 4-week-old plants or 0.1 g dry seeds were used for total lipid extraction in 20 ml chloroform/methanol/formic acid (1/2/0.1, by volume) for 1 h at room temperature by vigorous shaking. Then 10 ml of 1 M KCl-0.2 M $H_3PO_4$ was added. Following centrifugation at 10,000 g for 5 min, the lower organic fractions were transferred to new tubes, and 10 ml chloroform/methanol (1:2, by volume) was added to the remaining upper phase. After centrifugation, the lower organic phases were pooled and evaporated to dryness under $N_2$. Dried lipids were resuspended in 1 ml hexane and centrifuged for 5 min at 10,000 g. 0.5 ml lipids were spotted on a TLC plate and separated with dichloromethane as a developing solvent. TLC plates were spray-stained with ethanolic Berberine sulfate solution (0.01%, w/v) and free sterols were visualized by fluorescence ($A = 366$ nm). Bands containing free sterol (RF = 0.2–0.3) and steryl-ester (Rf = 0.9) fractions were scraped off. Free sterols were extracted with 3 ml of chloroform/methanol/formic acid (1/2/0.1, by volume) in the presence of 5 μg 5-α-Cholestane as an internal standard. Then, 1.5 ml 1 M KCl-0.2 M $H_3PO_4$ was added to the extract and the mixture was vortexed vigorously. After 5 min of centrifugation at 10,000 g, the organic fractions were transferred to new glass tubes, dried under $N_2$. Steryl-ester fractions were heated for 1 h at 90 °C with 6% potassium hydroxide in ethanol and the liberated free sterols were extracted with hexane. The sterols were then subjected to silylation in 100 μl N-Methyl-N-(tri-methylsilyl) trifluoroacetamide (MSTFA) for 2 h at 70 °C. After the reagents were evaporated under $N_2$, the sterol derivatives were dissolved in hexane and used for GC-MS analyses. The analysis of sterol derivatives was performed on an HP5975 gas chromatography–mass spectrometer (Hewlett–Packard) fitted with a J&W DB-5ms capillary column (30 m long, 0.25 mm inner diameter, film thickness 0.25 μm) with helium as a carrier gas. The temperature program used was 170 °C for 1.5 min, a fast increase from 170 to 280 °C and a slow increase from 280 to 300 °C. The inlet temperature was 280 °C. The sterols were identified based on their characteristic retention times and on well-documented fragmentation patterns of their tri-methylsilyl derivatives[63].

**RNA extraction and quantitative reverse transcription PCR (qRT-PCR)**. Total RNAs were extracted from mature leaves of 4-week-old plants using the TRIzol reagent (Thermo Fisher Scientific, cat# 15596026). First-strand cDNA was synthesized using M-MLV reverse transcriptase (New England Biolabs, cat# M0253). qRT-PCR was performed with SsoAdvanced Universal SYBR Green Supermix (Bio-Rad, cat# 1725270). The results were normalized to expression levels of *UBQ5*. The primers used for the qRT-PCR analysis are listed in Supplementary Table 4.

**Oleic acid feeding**. Oleic acid was dispersed and suspended in an incubation buffer (20 mM MES, pH 5.5, 0.01% Tween 20, one-tenth strength of MS salts, 0.5 mM oleic acid) by sonication for 15 s. Detached mature leaves of 5-week-old plants were then floated in the incubation buffer in the light (60 μmol m$^{-2}$ s$^{-1}$) at room temperature with gentle shaking for 12 h. The leaves were washed with water for four times (water 5 min each) and incubated in the same medium lacking oleic acid for 24 h. Total lipids were extracted and separated by TLC as described above. The TAG bands were visualized by brief exposure to iodine vapor and TAG was quantified by GC-MS.

**In vivo acetate and oleic acid pulse-chase labeling**. In vivo $^{14}$C-acetate pulse-chase assay was carried out according to Fan et al.[64]. Briefly, rapidly growing leaves of 5-week-old plants were detached and labeled for 1 h with $^{14}$C-acetate. After washing three times with water, the leaves were incubated further with unlabeled solution (20 mM MES, pH 5.5, one-tenth strength of MS salts, and 0.01% (v/v) Tween 20) under a 16-h-light/8-h-dark cycle for 16 h to 36 h. Total lipids were extracted and separated by TLC as described above. Label incorporation into total lipids was determined by liquid scintillation counting. The $^{14}$C-labeled TAG was separated by TLC and analyzed by phosphorimaging using a Typhoon FLA 7000 imager (GE Healthcare). Relative radioactivity of the TAG bands was quantified using the ImageQuant5.2 software.

For $^{14}$C-oleic acid pulse-chase labeling assay, mature leaves of 5-week-old plants were incubated with 0.02 mM $^{14}$C-oleic acid (50 mCi/mM, American Radiolabeled Chemicals) and then chase for 24 h as described above. Total lipids were extracted and separated by TLC. The radiolabeled TAG bands were visualized by brief exposure to iodine vapor and the radioactivity in TAG was quantified by liquid scintillation counting.

**LD isolation, LD size measurements, and LD protein analysis**. Seed LDs were isolated from Arabidopsis seed following the protocol from Ding et al.[65] with some modifications. Briefly, 40 mg dry seeds were ground in 5 mL of ice-cold buffer A (20 mM tricine, pH 7.8, 250 mM sucrose, 10 mM KCl, 1 mM EDTA, 1 mM MgCl$_2$, 5 mM β-mercaptoethanol, and 1 mM PMSF, 1 × cocktail) using a mortar and pestle. The crude homogenate was centrifuged at 3000 × g for 10 min and the supernatant (named postnuclear supernatant, PNS) was transferred to a 15 ml glass tube. Then 8 ml of buffer B (20 mM Hepes, pH 7.4, 100 mM KCl, 1 mM EDTA, 2 mM MgCl$_2$, 5 mM β-mercaptoethanol, and 1 mM PMSF, 1 × cocktail) was loaded on top of the PNS. Following centrifugation at 10,200 g using HB-6 Swinging-Bucket Rotor for 1 h at 4 °C, the LDs from the top LD fraction were carefully collected, suspended them in 1 ml buffer B, and then centrifuged at 17,000 g for 5 min at 4 °C. The LD fraction was suspended in 200 μl buffer B and used for further analysis.

To examine LD morphology, LDs were stained with Nile red, and viewed using a Zeiss epifluorescence microscope (Carl Zeiss; Axiovert 200 M) with a red fluorescent protein filter. The size and number of LDs were quantified using ImageJ software.

For TAG quantification and protein analysis of LDs, 1 μl of well suspended LDs were added to 60 μl chloroform to extract the lipids and the lipids were separated by TLC. Relative contents of TAG in 1 μl LDs from different genotypes were quantified using ImageJ software. LDs containing a similar amount of TAG were delipidated in 0.5 ml chloroform/acetone (1:1, v/v). After vortex and centrifugation (10000 g × 5 min), 400 μl lipids in the organic phase were transferred to a new tube and TAG concentration was measured by GC-MS. The leftover lipids, upper phase and pellet were air-dried and dissolved in 2 × laemmli sample buffer. Proteins from LDs containing 15 μg of TAG were separated by 15% SDS-PAGE gel, followed by staining with Coomassie blue. The gel was visualized using ImageQuant LAS 4000 biomolecular imager (GE Healthcare Life Sciences), and relative protein levels were quantified using ImageQuant TL software according to the manual.

**PDAT1 activity assay**. Microsomal membranes were prepared according to Xu et al.[66]. In general, about 1.5 g mature leaves from 5-week-old plants were powdered with liquid nitrogen in a mortar and pestle. A grinding medium (100 mM HEPES-KOH, pH 7.4, containing 1 mM EDTA, 0.32 M sucrose, and 1 mM dithiothreitol) was then added. The homogenates were filtered through two layers of Miracloth (Calbiochem, Gibbstown, NJ, USA), and centrifuged at 3000 g for 5 min, and the supernatant re-centrifuged at 15,000 g for 15 min. The supernatant was re-centrifuged at 120,000 g for 1 h and the resulting pellet was resuspended in grinding medium, sonicated for 30 s and protein concentrations were determined using Pierce BCA Protein Assay Kit (Thermo-Scientific, cat#23227). To prepare $^{14}$C-labeled PC[64], two-week-old seedlings were incubated in 20 mM MES-KOH, pH 6.0, with 0.2 mCi of $^{14}$C-acetate (American Radiolabeled Chemicals) for overnight. Total lipids were extracted and separated by TLC. $^{14}$C-labeled PC was eluted from silica gel using chloroform: methanol: formic acid (1:2:0.1, by volume) and redissolved in chloroform.

PDAT1 activity assay was conducted in a reaction mixture (100 μl) containing 100 mM HEPES-KOH (pH 7.4), 250 μM $^{14}$C-labeled PC and 250 μM 18:1-DAG (Avanti Polar Lipids), 0.5 mM ATP, 0.5 mM Coenzyme A, 1 mM MgCl$_2$, 0.02% Tween 20. The reaction mixture was thoroughly mixed and incubated at 30 °C for

30 min. The $^{14}$C-labelled TAGs were isolated by TLC and the radioactivity was determined by scintillation counting.

**Immunoblotting**. Embryos were dissected under dissecting microscope. Fifty embryos at the bent cotyledon stage were grounded with liquid N in the protein extraction buffer (50 mM HEPES-KOH, pH 7.6, 150 mM NaCl, 1 mM EDTA, 10% glycerol, 1 mM β-mercaptoethanol, 0.25% Triton-X 100, and 1 × complete protease inhibitor cocktail). Protein content was quantified using pierce BCA protein assay kit (ThermoFisher, cat#23227) according to the manufacturer's instructions. Proteins were separated by 12% SDS-PAGE gel and blotted to a PVDF membrane. OLE1-GFP protein was detected using mouse anti-GFP antibody (1:3000, Clontech, cat# 632380), and then with goat antimouse IgG HRP-conjugated secondary antibodies. HRP activity was detected using SuperSignal western detection reagents (Thermo Fisher Scientific) and visualized with an ImageQuant LAS 4000 biomolecular imager (GE Healthcare Life Sciences). The band intensity was measured using ImageJ software.

**Reporting Summary**. Further information on research design is available in the Nature Research Reporting Summary linked to this article.

## Data availability
Data supporting the finding of this work are available within the paper and its supplementary information files. Raw GC-MS data generated for quantification of TAGs and sterols are available upon reasonable request. Whole-genome sequencing data used during positional cloning are deposited in NCBI's SRA (Sequence Read Archive) under the accession number PRJNA760508. Source data are provided with this paper.

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

## Acknowledgements
This work was supported by the U.S. Department of Energy, Office of Science, Office of Basic Energy Sciences under contract number DE-SC0012704—specifically through the Physical Biosciences program of the Chemical Sciences, Geosciences and Biosciences Division. Use of the transmission electron microscope and the confocal microscope at the Center of Functional Nanomaterials was supported by the Office of Basic Energy Sciences, U.S. Department of Energy (DE-SC0012704).

## Author contributions
C.X. and J.F. designed the study. L.Y., J.F. and C.Z. performed the experiments. C.X. and L.Y. drafted the manuscript, and L.Y., J.F. and C.X. revised the manuscript.

## Competing interests
The authors declare no competing interests.
