## [Peer Review File · Nature Communications]

Sterols are required for the coordinated assembly of lipid droplets in developing seedsREVIEWER COMMENTS

Reviewer #1 (Remarks to the Author):

Aside from some minor typos and suggestions indicated in the attached PDF, this is a well written and well documented paper describing the role of a specific plant sterol intermediate in the formation of plant lipid droplets in leaves (synthetic system) and seeds. The combination of genetic and biochemical analyses is excellent and fully convinces me that this particular sterol and not others is necessary for the observed effects on lipid droplets. I do have a question about the mechanism that the authors may be able to address:

1. Change in gene expression of oleosin genes affected by the sterol could explain the observed phenotypes. This possibility is noted by the authors as well but argued. I think it should be readily feasible to determine RNA abundance for oleosin genes to rule out effects on gene expression or RNA stability.

2. The authors' favorite hypothesis is that sterols disturb the lipid droplet surface normally enabling integration of oleosins. I am wondering whether binding of the sterol to oleosin can be tested in lipid overlay assays or in a lipid droplet system by mixing TAG, PC and the sterol. This would require the production of oleosin protein in some recombinant system in its native form, which I do not know whether this can be done.

Reviewer #2 (Remarks to the Author):

The manuscript by Linhui Yu et al. entitled "Sterols are required for the coordinated assembly of lipid droplets in developing seeds" describes a lipid cellular phenotype of a series of Arabidopsis sterol mutants that carry loss of function mutations in genes encoding sterol biosynthetic enzymes. These mutants are viable dwarfs impaired in the conversion of sterol biosynthetic intermediates into the final products, of which campesterol (a 24-methylsterol) and sitosterol (a 24-ethylsterol) are the major ones. Campesterol is a precursor of brassinosteroids, and consequently the lack of campesterol turns down the production of brassinosteroids, therefore the dwarfism. This has been established in several papers quoted by the authors (these dwarfs have been picked up in collections of T-DNA insertional mutants of Arabidopsis thaliana and characterized previously by several groups).

Here the authors have carried out an elegant genetic screen searching for lipid droplet cellular phenotypes in a strong oleosin background as a reporter line. In this line, the strong expression of the lipid droplet structural protein oleosin enhances the number of cytoplasmic lipid droplets. A line M1-7 was shown to carry a recessive allele of dwarf5, a result which prompted the authors to compare lipid droplets and triacylglycerol formation in the series of dwarf mutants impaired in phytosterol biosynthesis as mentioned above. As a control line, a dwarf4 mutant that is wild-type for the phytosterol pathway but shows a deficiency in the synthesis of brassinosteroids was considered.

The authors propose to demonstrate that defects in two sterol biosynthetic enzymes, namely, sterol-5(6)-desaturase (DWARF7) and sterol-7-reductase (DWARF5), result in the absence of proper lipid droplet biogenesis in the strong oleosin background. Conversely, a defect in another sterol biosynthetic enzyme, namely, sterol- Δ 24-reductase (DWARF1), does not alter the lipid droplet production. The control dwarf4 line shows also an unaffected lipid droplet biogenesis.

The authors conclude that the deficiency in the production of 24-ethyl- Δ 5-sterols in dwarf5 and dwarf7 is responsible for a reduced TAG accumulation in lipid droplets.

In fact, dwarf1 has almost no 24-ethylsterols (but 24-ethylidenesterols like isofucoesterol) and consequently this fits with the criteria of such a deficiency causing reduced lipid droplet formation, but dwarf1 has a normal lipid droplet biogenesis (shown in Fig.3). Likewise, the lipid droplet defect is not as strong in dwarf7 (which has Δ 7-sterols) as in dwarf5 (which has Δ 5,7-sterols). This is shown in a Fig.7 and in a suppl.Fig.10, which is not reinforcing the conclusion.

The careful examination of the sterol compositions given in a series of supplemental tables but unfortunately not in the main text, let me conclude that dwarf5 is rather forming itself a category, because it has the strongest decrease in sterol amounts (Table S2, last line 'total'). In fact, all lines analyzed have an amount of sterols ranging between 37 and 70 mg/g, whereas dwarf5 has values ranging between 12-15 mg/g. I found this as a very striking difference.

A series of important data are missing to conclude about a causal link between sterol amounts or specific sterols, and lipid droplet production. To the best of my opinion, the amount of sterols in young, mature, and old leaves, which all differ by a lipid droplet production, and also the amount of free sterols in these membrane fractions are not given. The latter is mandatory to conclude on a causal relationship between 24-ethyl- Δ 5-sterols and the capacity to form lipid droplets containing triacylglycerols, and from this to formulate a working hypothesis on the recruitment of specific sterols for the lipid droplet biogenesis process that occurs at the ER of cells.

Figures

Fig.1. Panel b shows populations of lipid droplets in young, mature and old leaves from 4-week old plants and panel a shows these 4-week old plants.

How are the stage defined, and what leaves are taken from the rosettes shown in a ?

(is it possible to point these leaves with arrows ?)

The same comment applies to Fig.2.c.; Fig.3b.

Fig.2. While col-0/OLE1 shows a high amount of TAGs, old col leaves do not exhibit any TAG increase. Here, how is 'old' defined ? There are in fact reports (see for instance Brocard et al. *Front Plant Sci.* 2017, 8:894; Shimada et al. *Plant Signal Behav.* 2015, 10(2):e989036) indicating that old leaves accumulates lipid droplets and TAGs. I am surprised to see not even a slight increase in lipid droplets in col-0 (wild-type). How is this explained ?

In panel d, the level of TAGs is shown for the wild-type col, but the corresponding pictures of lipid droplets in col plants in a panel b and moreover in a panel c, is lacking. The visualization of lipid droplets in panel c is based on GFP fluorescence, but since lipophilic dyes like Nile red or Bodipy are used in other experiments presented in the paper like in Fig.4, it would be informative to see amounts of lipid droplets in col old leaves, which show less TAGs (panel d) than other OLE1 lines.

Col shows also a rather strong band close to the middle of the TLC, which is also strong for old leaves of all lines. Is this band identified and could it be relevant to understand the dwarf5/OLE1 phenotype ?

Fig.3. Loss of function alleles of genes encoding CVP1/SMT3 (sterol-C24-methyltransferases) or DWARF7 (sterol-C5(6)-reductase) but not DWARF1 (sterol-D24-reductase) recapitulate to a certain extent the M1-7 phenotype of low LD accumulation in the OLE1 background (partially in the case of *cvp1/smt3*).

To clearly understand the absence of causal relation between dwarfism due to a brassinosteroid deficiency and the low LD accumulation in dwarf sterol biochemical mutants (*dwarf5*, *dwarf7* but not *dwarf1*), which also lack brassinosteroids, it is important to show also the data for *dwarf4*, a biochemical mutant impaired in the conversion of sterols to brassinosteroids (the data appears in a Fig.S4, which could stand only with the data for the brassinosteroid signaling mutant *bri1*, which has wild-type sterol and wild-type brassinosteroid biosynthetic pathways).

Fig.4. The oleic acid feeding experiment is not shown in the case of *dwarf1*, which then would be expected by the reader to behave like a *dwarf4*. Is this the case ? Similarly, the confocal micrographs for *dwarf7* are not displayed in a panel c. Is there any specific reason, like data redundancy ? Is it assumed that what is true for *dwarf5* is also for *dwarf7* ? Apparently the results shown in a Fig.7 tells that *dwarf7* is closer to *dwarf1* than to *dwarf5*.

Supplemental Figures.

The legend of Fig.S1 should also mention the enzymes for the BR specific pathway.

Supplemental Fig.10. dwarf7/OLE1 cotyledons and roots are more close to dwarf1/OLE1 or col/OLE1 than to dwarf5/OLE1. Compared to what is shown in Fig.3 in the case of leaves, it looks like a very different situation. How to explain this ?

Other points, minor points, editing.

Introduction

Rephrase ‘.. the first parental sterol composed of a tetracyclic ring nucleus with a free hydroxyl group and side chain at the 3rd or 24th carbon atom, respectively.’ using appropriate carbon numbering (C-3, C-24)

In the sentence ‘Cycloartenol is the common substrate for the synthesis of cholesterol, a minor sterol in plants and of the major sterols’, define major sterols.

Results

Nomenclature throughout

Δ 5,7-stigmasterol: use stigmasta-5,7,22-trienol

Δ 5,7-sitosterol: stigmasta-5,7-dienol

sterol Δ 7 C-5 desaturase: use Δ 7-sterol-C5-desaturase

sterol Δ 7 C-5 desaturase use Δ 7 -sterol-C5-desaturase

Δ 5-24-methylsterols, Δ 5-24-ethylsterols: 24-methyl- Δ 5-sterols, 24-ethyl- Δ 5-sterols

Legend for Fig.6. line 3: wild-type, not wide type

Methods

In the section “Sterol measurements by GC-MS”, line 10 from bottom paragraph: The sterols were then subjected to acetylation in 100 μ l N-Methyl-N-(trimethylsilyl) trifluoroacetamide (MSTFA) for 2 h at 70°C:

The incubation of sterols with MSTFA is a silylation reaction, not an acetylation. Which reaction was performed ?

RESPONSE TO REVIEWER COMMENTS

Reviewer #1 (Remarks to the Author):

Aside from some minor typos and suggestions indicated in the attached PDF, this is a well written and well documented paper describing the role of a specific plant sterol intermediate in the formation of plant lipid droplets in leaves (synthetic system) and seeds. The combination of genetic and biochemical analyses is excellent and fully convinces me that this particular sterol and not others is necessary for the observed effects on lipid droplets. I do have a question about the mechanism that the authors may be able to address:

1. Change in gene expression of oleosin genes affected by the sterol could explain the observed phenotypes. This possibility is noted by the authors as well but argued. I think it should be readily feasible to determine RNA abundance for oleosin genes to rule out effects on gene expression or RNA stability.

RESPONSE: Thank you very much for reviewing our manuscript carefully, providing constructive comments and suggestions and correcting typos and grammar errors. Data on changes in *OLE1* transcript level in sterol mutants, two other LD-deficient mutants (*M2-1* and *M3-1*) and the *PDAT1* knockout mutant in the *OLE* line background (*pdat1/OLE1*) are now shown in Supplemental Fig. 6. Both *M2-1* and *M3-1* were isolated from the same genetic screen as for *dwf5-10/OLE1* but they are not dwarf and grow well on soil like the wild type, suggesting they are not deficient in sterols. *PDAT1* is the key enzymes in oil synthesis in leaves (Fan et al., Plant Cell 2013, 25:3506-) and knockout of *PDAT1* results in a drastic decrease in LD accumulation in leaves of the *OLE1* line, and again *pdat1/OLE1* presumably contains normal sterol levels because it grows similarly to the wild type. Results on analysis of *OLE1* gene expression in these mutants indicate that 1) the *OLE1* transcript level is drastically decreased in all LD deficient mutants; 2) compared with young leaves, old leaves have a much higher *OLE1* transcript level in all LD-deficient mutants; and 3) there is a strong positive correlation between LD number and *OLE1* transcript abundance. A close coupling between LD abundance and levels of oleosin transcripts and proteins has also been noted in many studies in seeds (Ruuska et al., Plant Cell 2002, 14: 1191-; Miquel et al., Plant Physiol 2014, 164: 1866-; Hseih et al. Plant Physiol. 2004, 136: 3427-). Disruption of *WRINKLED1*, a key transcription factor controlling fatty acid synthesis gene expression but not oleosin gene expression, results in drastic decreases in accumulation of both oleosin transcripts and oil in seeds (Maeo et al., Plant J 2009, 60: 476-). In addition, both TAG content and oleosin transcript decrease in seeds of the Arabidopsis mutant defective in diacylglycerol acyltransferase1 (Zou et al., Plant Mol Biol 1996, 31:429-). These results strongly suggest that the *OLE1* transcript level is controlled by LD abundance but not by sterol levels. However, sterols likely exert their effects on *OLE1* transcripts via their effects on LD abundance. Sterol deficiency results in a decrease in LD abundance, and hence a decrease in *OLE1* transcript levels. We discuss these important points in the first paragraph of Discussion as follows:

“In this study, a comprehensive genetic, biochemical and cell biological analysis uncovers a BR-independent role of sterols in LD accumulation in plants. Sterols can function as signaling molecules in regulating gene transcription in plants⁶. however, our analysis revealed a strong positive correlation between the *OLE1* transcript level and LD abundance in *dwf* mutants and other mutants with normal

growth phenotypes. Similarly, TAG and LD accumulation has often been found to concur with increases in oleosin protein and transcript levels in developing seeds. Disruption of *WRINKLED1*, a key transcription factor controlling genes involved in synthesis of fatty acids but not oleosins, results in drastic decreases in levels of both TAGs and oleosin transcripts in seeds³⁷. In addition, TAG content and oleosin expression have been found to decrease concomitantly in seeds of a mutant disrupted in TAG synthesis³⁸. Together, these results strongly suggest that oleosin transcript levels are controlled by LD abundance and imply that sterols impact oleosin gene expression indirectly via their effects on LD assembly.”

A close correlation between LD abundance and levels of oleosin protein and transcripts likely reflects the unique properties of oleosin proteins as discussed in Discussion section, which reads “Due to their long hydrophobic stretch, oleosins are predicted to be unstable when present in ER membranes and become stable when associated with LDs²⁵. Likely for this reason, the transcript levels of genes involved in oil and oleosin synthesis are coordinately regulated during seed development⁴⁰. In addition, oleosins are synthesized in ER domains alongside with the synthesis of TAGs⁴¹ and the assembly of nascent LDs⁴². Presumably, this spatiotemporal regulation enables a concomitant accumulation of TAG and oleosins and efficient transfer of oleosins to LDs^{25, 41, 42} during oil accumulation.” Therefore, lack of sufficient LDs would result in a rapid degradation of oleosins, which may invoke a feedback signal leading to a rapid degradation of oleosin transcripts or to suppression of oleosin gene expression. The latter possibility, however, is rather remote since the *OLE1* gene used in our genetic screen lacks regulatory sequences and is driven by a 35S promoter.

Regulation of RNA stability is a major mechanism controlling mRNA levels in diverse eukaryotic systems. However, addressing the mechanism underlying the regulation of *OLE1* transcript stability requires an entire series of new quantitative experiments and is beyond the scope of this work, since the *OLE1* transcript level correlates with LD abundance, and this manuscript, however, focuses on role of sterols in LD biogenesis.

Results on *OLE1* expression profiles are now described in Results with a subtitle “The *OLE1* transcript level correlates strongly with LD abundance”, which now reads: “To test whether sterols directly impact LD abundance via their effects on *OLE1* expression, we analyzed the *OLE1* transcript level in *dwf* mutants, two additional LD-deficient mutants (*M2-1* and *M3-1*) isolated from the same genetic screen as for *M1-7* and the phospholipid:diacylglycerol acyltransferase1 (*PDAT1*) knockout line carrying the *OLE1* transgene (*pdat1-2/OLE1*). Unlike sterol-deficient *dwf* mutants, *M2-1*, *M3-1* and *pdat1-2/OLE1* showed normal growth phenotypes and thus presumably contained normal levels of sterols. However, like *dwf5* and *dwf7*, all these three LD-deficient mutants had drastic decreases in levels of the *OLE1* transcript and there is a strong positive correlation between *OLE1* transcript levels and LD abundance in leaves (Supplemental Fig. 6). These results suggest that the expression level of the *OLE1* transgene is related to LD abundance rather than to amounts of sterols per se.”

2. The authors' favorite hypothesis is that sterols disturb the lipid droplet surface normally enabling integration of oleosins. I am wondering whether binding of the sterol to oleosin can be tested in lipid overlay assays or in a lipid droplet system by mixing TAG, PC and the sterol. This would require the production of oleosin protein in some recombinant system in its native form, which I do not know whether this can be done.

RESPONSE: We appreciate this comment and the suggestions by the reviewer. Yes, one of the main problems associated with the proposed experiments is that oleosins are insoluble in water (Li et al., *JBC* 1992,267: 8245-; Roux et al., *J Agric Food Chem* 2004, 52: 5245-; Kim et al., *Biotech Bioproc Eng* 2007, 12: 542-). In our hands, OLE1 was expressed well in *E. coli* but it was not soluble and affinity purification yielded no recombinant proteins despite extensive research efforts (please see Fig. A and B below). We also tried a different approach. Here we first purified LDs from seeds by float centrifugation. LDs were then delipidated and LD proteins were solubilized using a mild detergent NP40 (1.5%). Oleosins solubilized from seed LDs were then used for lipid-protein overlay assays. As a positive control, a small aliquot of solubilized oleosin proteins was spotted on the membrane. Western blot analysis using OLE1 antibody failed to detect binding of oleosins with cholesterol, sitosterol and stigmasterol and campesterol. Weak signals were detected with stigmasterol, but they are likely to be background noise since the signal intensity did not increase with increasing amounts of spotted stigmasterol (please see Fig. C and D below). Because of the uncertainties about the overlay results and because we do not know whether the solubilized oleosins maintain their native forms and because of the article length limits, we opted to not include these lipid-protein overlay results in the manuscript.

A-B. SDS-PAGE analysis of the His-tagged OLE1 protein from *E. coli* cell lysate before induction, after induction, in soluble and pellet fractions and during elution.
C. Purified seed lipid droplet proteins. Lipid droplets were purified using a float centrifugation method. LDs were suspended in 200 μ l buffer and delipidated by extracting with 5 volumes diethyl ether for 3 times. The ether-water interface layer was collected, dry under nitrogen, then 20 mM Tris-HCl (pH7.4, with 1.5% NP40) was added to solubilized oleosin proteins.
D. Sterols-oleosin overlay assay. Ten pmol and 100 pmol individual sterols in chloroform were loaded to nitrocellulose membrane and air dried. One μ g seed LD proteins were loaded as a positive control. Chloroform was loaded as a blank control. The membrane was blocked in 3% BSA (without fatty acid) for overnight, then incubated with solubilized LD proteins (2 μ g/ml) in 3% BSA for 2 h. After extensive washes, the membrane was incubated with anti-OLE1 antibody and secondary antibody sequentially. HRP activity was detected using SuperSignal western detection reagents.

Reviewer #2 (Remarks to the Author):

The manuscript by Linhui Yu et al. entitled "Sterols are required for the coordinated assembly of lipid droplets in developing seeds" describes a lipid cellular phenotype of a series of Arabidopsis sterol mutants that carry loss of function mutations in genes encoding sterol biosynthetic enzymes. These mutants are viable dwarfs impaired in the conversion of sterol biosynthetic intermediates into the final products, of which campesterol (a 24-methylsterol) and sitosterol (a 24-ethylsterol) are the major ones. Campesterol is a precursor of brassinosteroids, and consequently the lack of campesterol turns down the production of brassinosteroids, therefore the dwarfism. This has been established in several papers quoted by the authors (these dwarfs have been picked up in collections of T-DNA insertional mutants of Arabidopsis thaliana and characterized previously by several groups).

Here the authors have carried out an elegant genetic screen searching for lipid droplet cellular phenotypes in a strong oleosin background as a reporter line. In this line, the strong expression of the lipid droplet structural protein oleosin enhances the number of cytoplasmic lipid droplets. A line M1-7 was shown to carry a recessive allele of dwarf5, a result which prompted the authors to compare lipid droplets and triacylglycerol formation in the series of dwarf mutants impaired in phytosterol biosynthesis as mentioned above. As a control line, a dwarf4 mutant that is wild-type for the phytosterol pathway but shows a deficiency in the synthesis of brassinosteroids was considered.

The authors propose to demonstrate that defects in two sterol biosynthetic enzymes, namely, sterol-5(6)-desaturase (DWARF7) and sterol-7-reductase (DWARF5), result in the absence of proper lipid droplet biogenesis in the strong oleosin background. Conversely, a defect in another sterol biosynthetic enzyme, namely, sterol- Δ 24-reductase (DWARF1), does not alter the lipid droplet production. The control dwarf4 line shows also an unaffected lipid droplet biogenesis.

The authors conclude that the deficiency in the production of 24-ethyl- Δ 5-sterols in dwarf5 and dwarf7 is responsible for a reduced TAG accumulation in lipid droplets.

In fact, dwarf1 has almost no 24-ethylsterols (but 24-ethylidenesterols like isofucosterol) and consequently this fits with the criteria of such a deficiency causing reduced lipid droplet formation, but dwarf1 has a normal lipid droplet biogenesis (shown in Fig.3).

RESPONSE: Thank you very much for reviewing our manuscript carefully, providing valuable comments and suggestions and bring our attention to this very important point regarding dwarf1. We are very sorry that we mistakenly regarded isofucosterol as a 24-ethyl- Δ 5-sterol. In the revised manuscript, we have changed our description from "deficiency in 24-ethyl- Δ 5-sterols" to "deficiency

in 24-ethyl- Δ^5 -sterols and 24-ethylidene- Δ^5 -sterol". We also discussed the structural and functional similarity between 24-ethyl- Δ^5 -sterols and 24-ethylidene- Δ^5 -sterol in the Discussion, which reads: "disruption of DWF1 causes drastic decreases in sitosterol and campesterol with concomitant increases in isofucoesterol and 24-methylenecholesterol but no significant effects on total Δ^5 -sterol content and LD formation. These results suggest that isofucoesterol and 24-methylenecholesterol are functionally interchangeable with sitosterol and campesterol, respectively in LD biogenesis in plants. In fact, isofucoesterol and 24-methylenecholesterol differ structurally from sitosterol and campesterol only by a single double bond at C-24 of the side chains and the key structural features required for optimal interaction with phospholipids are present in both types of sterols."

Likewise, the lipid droplet defect is not as strong in dwarf7 (which has Δ^7 -sterols) as in dwarf5 (which has $\Delta^5,7$ -sterols). This is shown in a Fig.7 and in a suppl.Fig.10, which is not reinforcing the conclusion.

RESPONSE: Thank you for the comments. We agree that *dwf7* has a weak LD deficient phenotype compared with *dwf5*, particularly in seeds. We consider this is possibly because *dwf7* contains more 24-ethyl- Δ^5 -sterols than *dwf5* in seeds and young seedlings. We believe that although there are notable differences in LD phenotypes between these two mutants, the results do support our conclusion that levels of 24-ethyl- Δ^5 -sterols and 24-ethylidene- Δ^5 -sterol are important for proper LD biogenesis. We discuss this important point in the Discussion, which reads: "In addition to disturbed sterol composition, changes in overall sterol content appear also to impact LD biogenesis in plants. This notion is supported by several lines of evidence. First, total free sterol levels are reduced only in *dwf5* mutants severely deficient in LD biogenesis. Second, among all the sterol mutants analyzed, *dwf5* has the lowest level of free sterols and this mutant exhibits the most severe LD deficiency phenotype in both leaves and seeds. Third, leaf senescence is known to be associated with an increased accumulation of sterols¹³ and TAG levels, LD number (Fig. 1-3) and total sterol content (Supplemental Fig. 13) are all increased in old leaves of both *dwf5* and *dwf7*. Fourth, *dwf5* and *dwf7* mutants contained higher levels of 24-ethyl- Δ^5 -sterols in seeds (Supplemental Table 4) than in leaves (Supplemental Table 2) and both mutants showed less severe defects in storage lipid accumulation in seeds than in leaves. Since most of the seed mass is storage lipids and proteins, it seems likely that actual levels of free sterols in cellular membranes of *dwf5* and *dwf7* mutants are much higher than those reported in Supplemental Table 4. Alternatively, the observed more pronounced defects in LD accumulation in leaves than in seeds may reflect a different mechanistic basis underlying LD biogenesis between these two organs."

The careful examination of the sterol compositions given in a series of supplemental tables but unfortunately not in the main text, let me conclude that dwarf5 is rather forming itself a category, because it has the strongest decrease in sterol amounts (Table S2, last line 'total'). In fact, all lines analyzed have an amount of sterols ranging between 37 and 70 mg/g, whereas dwarf5 has values ranging between 12-15 mg/g. I found this as a very striking difference.

A series of important data are missing to conclude about a causal link between sterol amounts or specific sterols, and lipid droplet production. To the best of my opinion, the amount of sterols in young, mature, and old leaves, which all differ by a lipid droplet production, and also the amount of free sterols in these membrane fractions are not given. The latter is mandatory to conclude on a causal relationship between 24-ethyl- Δ^5 -sterols and the capacity to form lipid droplets containing triacylglycerols, and from

this to formulate a working hypothesis on the recruitment of specific sterols for the lipid droplet biogenesis process that occurs at the ER of cells.

RESPONSE: Thank you for bringing this important point to our attention. Levels of sterols in leaves of different ages are now shown in Supplemental Fig. 13. Overall, the data indicate that there is an increase in sterol content as leaves age.

Figures

Fig.1. Panel b shows populations of lipid droplets in young, mature and old leaves from 4-week old plants and panel a shows these 4-week old plants.

How are the stage defined, and what leaves are taken from the rosettes shown in a ?
(is it possible to point these leaves with arrows ?)

The same comment applies to Fig.2.c.; Fig.3b.

RESPONSE: We appreciate this comment by the reviewer. The leaf developmental stages are defined and mentioned in legends of Fig. 1, 2 and 3 as follows: young leaves: the upper growing leaves; mature leaves: the newest expanded leaves; old leaves: the oldest expanded green leaves. Representative different developmental stages of leaves are now indicated with arrows in Fig. 2b.

Fig.2. While col-0/OLE1 shows a high amount of TAGs, old col leaves do not exhibit any TAG increase. Here, how is 'old' defined ? There are in fact reports (see for instance Brocard et al. Front Plant Sci. 2017, 8:894; Shimada et al. Plant Signal Behav; 2015, 10(2):e989036) indicating that old leaves accumulates lipid droplets and TAGs. I am surprised to see not even a slight increase in lipid droplets in col-0 (wild-type). How is this explained ?

RESPONSE: We appreciate this comment by the reviewer. The terms "old" refers to the developmental status of the leaf rather than to its chronological age. We used 4-week-old plants in our experiments. Under our growth conditions, the old leaves we used for TAG and sterol analysis were still fully green and thus are probably not at senescent stages. We noted that the truly senescent leaves in 4-week-old plants are just the first true leaves, which showed yellowing at the leaf tips (the typical indicator of leaf senescence). We did not use these senescent leaves for our experiments because these leaves are too small and it is very difficult, if not impossible, to obtain enough fresh leaves for TAG and sterol analyses, both of which are low abundant metabolites and accurately determining their levels requires large amounts of fresh tissues to start with. Therefore, the use of "old" rather than senescent leaves is probably the reason as to why we only detected relatively small increases in TAG content and LD number. It should also be noted that TAG content and LD abundance are not only influenced by developmental stages but also by a lot of other factors such as light intensity (Yu et al., Plant Physiol 2021, 185: 94-) and many other environmental cues (Lu et al., Plants, 2020, 9: 472). For this reason, increases in TAG content during leaf senescence have been observed in some experiments as pointed out by the reviewer but not in others (e. g. Yang et al., Plant Physiol 2009, 150: 1981-). Therefore, the second possible reason for low abundance of TAG and LDs in old leaves is because of differences in experimental conditions, in particular, the nutritional status.

In panel d, the level of TAGs is shown for the wild-type col, but the corresponding pictures of lipid droplets in col plants in a panel b and moreover in a panel c, is lacking. The visualization of lipid droplets in panel c is based on GFP fluorescence, but since lipophilic dyes like Nile red or Bodipy are used in other experiments presented in the paper like in Fig.4, it would be informative to see amounts of lipid droplets in col old leaves, which show less TAGs (panel d) than other OLE1 lines.

RESPONSE: We appreciate the suggestion. We have added the images taken from Nile red-stained wild-type leaves to Fig. 2c.

Col shows also a rather strong band close to the middle of the TLC, which is also strong for old leaves of all lines. Is this band identified and could it be relevant to understand the dwarf5/OLE1 phenotype ?

RESPONSE: We appreciate this comment by the reviewer. For reasons that we do not understand, the band shows up in some experiments but not in others even in lipid extracts from the same genotype grown under the same conditions (comparing Fig. 2d with Fig. 3b). This inconsistency discouraged us, at this time, from further analyzing the nature of the band, and suggests that the band is unlikely to be linked to the LD-deficient phenotype in the dwarf mutants.

The nature of the band remains unknown, and we are planning to investigate into it in future work.

Fig.3. Loss of function alleles of genes encoding CVP1/SMT3 (sterol-C24-methyltransferases) or DWARF7 (sterol-C5(6)-reductase) but not DWARF1 (sterol-D24-reductase) recapitulate to a certain extent the M1-7 phenotype of low LD accumulation in the OLE1 background (partially in the case of *cvp1/smt3*). To clearly understand the absence of causal relation between dwarfism due to a brassinosteroid deficiency and the low LD accumulation in dwarf sterol biochemical mutants (*dwarf5*, *dwarf7* but not *dwarf1*), which also lack brassinosteroids, it is important to show also the data for *dwarf4*, a biochemical mutant impaired in the conversion of sterols to brassinosteroids (the data appears in a Fig.S4, which could stand only with the data for the brassinosteroid signaling mutant *bri1*, which has wild-type sterol and wild-type brassinosteroid biosynthetic pathways.

RESPONSE: We appreciate the suggestion by the reviewer. The data for *dwf4* are now added to Fig. 3.

Fig.4. The oleic acid feeding experiment is not shown in the case of *dwarf1*, which then would be expected by the reader to behave like a *dwarf4*. Is this the case ? Similarly, the confocal micrographs for *dwarf7* are not displayed in a panel c. Is there any specific reason, like data redundancy ? Is it assumed that what is true for *dwarf5* is also for *dwarf7* ? Apparently the results shown in a Fig.7 tells that *dwarf7* is closer to *dwarf1* than to *dwarf5*.

RESPONSE: We appreciate the comments. Oleic acid feeding data for *dwf1* are now added to Supplemental Fig. 9. In addition, confocal images for *dwf7* have been included in Fig. 4c.

The results in Supplemental Fig. 9 indicate that *dwf1* behaved like the *dwf4* after oleic acid feeding and during the chase. TAG contents decreased significantly in both *dwf1* and *dwf4* while

maintained high levels in *dwf1/OLE1* and *dwf4/OLE1* after 24 h of chase. As showed in Fig. 4c, both *dwf5* and *dwf7* showed significantly decreased OLE1-GFP protein abundance compared with wild type or *dwf1*. However, we agree that the phenotypes in *dwf7* are less severe compared with those found in *dwf5* and we attribute these phenotypic differences to differences in sterol levels in these two mutants in seeds and seedlings and discussed this in the second to the last paragraph of the Discussion section.

Supplemental Figures.

The legend of Fig.S1 should also mention the enzymes for the BR specific pathway.

RESPONSE: The enzymes for the BR pathway are now mentioned in the Fig. S1 legend.

Supplemental Fig.10. *dwarf7/OLE1* cotyledons and roots are more close to *dwarf1/OLE1* or *col/OLE1* than to *dwarf5/OLE1*. Compared to what is shown in Fig.3 in the case of leaves, it looks like a very different situation. How to explain this ?

RESPONSE: We appreciate the comment. The images in Supplemental Fig. 10 (which is Supplemental Fig. 12 now) were obtained from 6-d-old young seedlings grown on agar plants, while images in Fig. 3 were obtained from leaves of 4-week-old plants grown on soil. The differences in developmental stages and growth conditions may account for the observed differences. The images in Supplemental Fig. 10 indicate that *dwf1* behaved like the wild type, while both *dwf5* and *dwf7* showed significantly decreased OLE1-GFP signals compared with wild type or *dwf1*. However, we agree that the phenotypes in *dwf7* are less severe compared with those found in *dwf5* and we attribute these phenotypic differences to differences in sterol levels in these two mutants. Compared with *dwf5*, *dwf7* contains higher levels of 24-ethyl- Δ^5 -sterols in seeds and presumably also in young seedlings.

Introduction

Rephrase ‘.. the first parental sterol composed of a tetracyclic ring nucleus with a free hydroxyl group and side chain at the 3rd or 24th carbon atom, respectively.’ using appropriate carbon numbering (C-3, C-24)

RESPONSE: Modified as suggested. Thank you.

In the sentence ‘Cycloartenol is the common substrate for the synthesis of cholesterol, a minor sterol in plants and of the major sterols’, define major sterols.

RESPONSE: Thank you for the suggestion. Plant major sterols are now defined as sitosterol, stigmasterol and campesterol.

Results

Nomenclature throughout

$\Delta^5,7$ -stigmasterol: use stigmasta-5,7,22-trienol

$\Delta^5,7$ -sitosterol: stigmasta-5,7-dienol

sterol Δ^7 C-5 desaturase: use Δ^7 -sterol-C5-desaturase

sterol Δ^7 C-5 desaturase use Δ^7 -sterol-C5-desaturase

Δ^5 -24-methylsterols, Δ^5 -24-ethylsterols: 24-methyl- Δ^5 -sterols, 24-ethyl- Δ^5 -sterols

Legend for Fig.6. line 3: wild-type, not wide type

RESPONSE: Modified as suggested. Thank you.

Methods

In the section "Sterol measurements by GC-MS", line 10 from bottom paragraph: The sterols were then subjected to acetylation in 100 μ l N-Methyl-N-(trimethylsilyl) trifluoroacetamide (MSTFA) for 2 h at 70°C:

The incubation of sterols with MSTFA is a silylation reaction, not an acetylation. Which reaction was performed ?

RESPONSE: Thank you for pointing this out. We use MSTFA for the derivatization. It should be a silylation reaction. We have corrected the error.

REVIEWERS' COMMENTS

Reviewer #1 (Remarks to the Author):

All my remaining questions have been adequately addressed by providing additional data and edits to the original manuscript. I see no further issues with this manuscript.

Reviewer #2 (Remarks to the Author):

I have read the revised version of the manuscript and revised Figures and Supplementals. All points of criticism have been addressed. In a supplemental Fig.6 appear now new biological materials Arabidopsis lines M2-1 and M3-1 also isolated in the genetic screen in addition to the M3-1/dwarf5 analyzed and discussed in the paper. Are these new lines genetically defined ?

REVIEWERS' COMMENTS

Reviewer #1 (Remarks to the Author):

All my remaining questions have been adequately addressed by providing additional data and edits to the original manuscript. I see no further issues with this manuscript.

Reviewer #2 (Remarks to the Author):

I have read the revised version of the manuscript and revised Figures and Supplementals. All points of criticism have been addressed. In a supplemental Fig.6 appear now new biological materials Arabidopsis lines M2-1 and M3-1 also isolated in the genetic screen in addition to the M3-1/dwarf5 analyzed and discussed in the paper. Are these new lines genetically defined ?

Response: Not yet. We have conducted whole genome sequencing of M2-1 and M3-1 and identified several candidate genes. We are now trying to figure out the exact mutations responsible for the lipid droplet-deficient phenotype.